# FREE LUNCH FOR STABILIZING RECTIFIED FLOW INVERSION

**Chenru Wang**
AGI Lab, Westlake University
wangchenru@westlake.edu.cn

**Beier Zhu**
University of Science and Technology of China
beier.zhu@ustc.edu.cn

**Chi Zhang**[*]
AGI Lab, Westlake University
chizhang@westlake.edu.cn

## ABSTRACT

Rectified-Flow (RF)-based generative models have recently emerged as strong alternatives to traditional diffusion models, demonstrating state-of-the-art performance across various tasks. By learning a continuous velocity field that transforms simple noise into complex data, RF-based models not only enable high-quality generation, but also support training-free inversion, which facilitates downstream tasks such as reconstruction and editing. However, existing inversion methods, such as vanilla RF-based inversion, suffer from approximation errors that accumulate across timesteps, leading to unstable velocity fields and degraded reconstruction and editing quality. To address this challenge, we propose Proximal-Mean Inversion (`PMI`), a training-free gradient correction method that stabilizes the velocity field by guiding it toward a running average of past velocities, constrained within a theoretically derived spherical Gaussian. Furthermore, we introduce `mimic-CFG`, a lightweight velocity correction scheme for editing tasks, which interpolates between the current velocity and its projection onto the historical average, balancing editing effectiveness and structural consistency. Extensive experiments on PIE-Bench demonstrate that our methods significantly improve inversion stability, image reconstruction quality, and editing fidelity, while reducing the required number of neural function evaluations. Our approach achieves state-of-the-art performance on the PIE-Bench with enhanced efficiency and theoretical soundness.

## 1 INTRODUCTION

Recently, Rectified-Flow (RF)-based generative models have gained increasing popularity over traditional diffusion models, achieving remarkable progress in a wide range of applications such as image synthesis (Labs et al., 2025; Esser et al., 2024), video generation (Wang et al., 2024; Li et al., 2025a; Zhao et al., 2025; Zhou et al., 2025), 3D content creation (Li et al., 2025c; Go et al., 2025; Li et al., 2024; Song et al., 2025), and other areas(Jin et al., 2025; Ma et al., 2026). These models operate by learning a continuous velocity field that transforms samples from a simple noise distribution (typically standard Gaussian) into complex data distributions. Beyond generation, RF-based models also offer the potential for inversion task, the process of mapping observed data samples back into their corresponding latent representations using a pre-trained RF-based model. This inversion capability opens up a range of downstream applications without requiring further training or fine-tuning of the model. For instance, high-quality inversion latent can support tasks such as image reconstruction and image editing (Deng et al., 2024; Rout et al., 2025; Wang et al., 2025a; Xu et al., 2025; Jiao et al., 2025; Ma et al., 2025).

Despite these potentials, achieving accurate and robust inversion remains a significant challenge, and the quality of inversion directly impacts the fidelity, consistency, and controllability of downstream

---

[*]Corresponding author.

tasks. A representative example is the DDIM inversion (Song et al., 2022), which has been widely adopted for inverting diffusion models (Pan et al., 2023; Samuel et al., 2025; Mokady et al., 2023; Han et al., 2024a; Kadosh et al., 2025). It uses a deterministic sampling equation that includes a neural network term to predict the noise at each timestep. Since the exact latent variable is inaccessible during inversion, the network input is typically approximated using the observed data at the current timestep. This approximation introduces estimation errors at each step, and these errors compound as the inversion progresses through multiple steps. As a result, the inversion latent representation deviates from the original noise prior (e.g., standard Gaussian) (Łukasz Staniszewski et al., 2025), which ultimately degrades the performance of tasks such as image reconstruction and editing. Similarly, the inversion of these RF-based models also introduces approximation errors, which affect the performance of downstream tasks.

To address the instability introduced by approximation errors in velocity fields, we aim to improve the reliability of inversion and editing by guiding the perturbed velocity field toward a more stable evolution. Existing research (Zhang et al., 2025) has found that such instability from the inherent sparsity of the generative distribution, which makes the approximation errors during inference difficult to eliminate entirely. These errors accumulate across timesteps, leading to disturbed velocity fields and ultimately degrading the stability of the inversion and reconstruction processes. To mitigate this issue, we propose Proximal-Mean Inversion (`PMI`), which performs gradient correction on the velocity field. At each timestep, `PMI` computes a running average of the predicted velocities from the initial step to the current one and applies a proximal update that guides the current velocity prediction toward this averaged direction. Furthermore, to prevent the corrected trajectory from deviating into low-density regions of the data distribution, we constrain each correction step within a spherical Gaussian, whose radius is theoretically derived in our work based on the instability analysis. This correction method stabilizes the inversion process and reduces the probability of falling into low-density regions, without additional training or neural function evaluations (NFEs), and can be seamlessly integrated into any RF-based model. `PMI` yields improved reconstruction quality and editing performance while maintaining sampling efficiency.

Beyond reconstruction, we identify that velocity perturbations also adversely affect editing tasks (Łukasz Staniszewski et al., 2025; Jiao et al., 2025), one of the most practical applications of inversion. Excessive perturbations to the velocity can degrade structural consistency, while velocities with insufficient deviation can compromise editing capabilities. To strike a balance between structural consistency and editing capabilities, we introduce `mimic-CFG`, a lightweight velocity correction strategy inspired by classifier-free guidance. Specifically, at each step, we compute the running average of the past velocities and project the current predicted velocity onto this average direction. The final corrected velocity is then obtained by interpolating between the predicted velocity and its projection. Experimental results on the PIE-Bench (Ju et al., 2024) show that our methods effectively correct velocity, balancing the two aforementioned factors, significantly reducing the number of sampling steps while enhancing editing quality and structural consistency.

We conduct extensive experiments across multiple benchmarks and tasks to evaluate the effectiveness of our methods. Results show that our methods consistently enhance baselines across multiple key metrics. Notably, by integrating our methods, many baselines can reach or exceed state-of-the-art (SOTA) quality with fewer NFEs, while preserving theoretical soundness and computational efficiency. Our contributions are summarized as follows:

- We propose `PMI`, a training-free method that corrects the perturbed velocity field via gradient correction toward a running average, effectively reducing inversion errors in RF-based models.

- We propose `mimic-CFG`, a lightweight velocity correction strategy that mirrors the structure of Classifier-Free Guidance (CFG) (Ho & Salimans, 2022) by interpolating between the current velocity and its projection onto the running average direction.

- Our approach establishes a new SOTA on the PIE-Bench, achieving superior editing and reconstruction performance with fewer sampling steps and no additional NFE.

## 2 RELATED WORK

**Inversion.** DDIM inversion aims to map real data to noise that approximates standard Gaussian. However, diffusion model's approximations at each timestep introduce errors, causing deviations

from the ideal noise Zhu et al. (2025a); Wang et al. (2025c;b); Zhu et al. (2025b). To address this, various methods have been proposed (Mokady et al., 2023; Dong et al., 2023a; Cho et al., 2024; Han et al., 2024b; Miyake et al., 2025; Wallace et al., 2022; Huberman-Spiegelglas et al., 2024), including Fixed-Point Iteration (FPI), which iteratively refines predictions to reduce local errors (Pan et al., 2023; Bao et al., 2025; Samuel et al., 2025; Garibi et al., 2025; Meiri et al., 2023), yielding more accurate noise estimates.

With the rise of flow-based models, inversion has shifted from solving stochastic differential equations (SDEs) to ordinary differential equations (ODEs). Building on this evolution, several methods have been proposed: RF-Inversion (Rout et al., 2025) uses dynamic optimal control to reduce inversion errors; RF-Solver (Wang et al., 2025a) employs high-order Taylor expansions to minimize integration errors; FireFlow (Deng et al., 2024) achieves second-order accuracy with only one NFE per step; and Two-stage Inversion (Xu et al., 2025) applies multiple fixed-point iterations per step to further refine predictions. While these approaches improve accuracy, especially those using FPI, they typically increase sampling steps and NFEs, leading to higher computational cost. To overcome this, we propose `PMI`, which reduces inversion error while requiring fewer sampling steps.

**Editing.** Image editing requires conditionally modifying semantic content while preserving original structure. Diffusion-based methods typically follow two steps: (1) invert the input image to its latent noise; (2) generate the edited image via guided sampling under editing constraints. However, inversion errors often degrade control and final image quality. To improve editing accuracy, some efforts introduce additional guidance, such as masks or image features. For instance, Prompt-to-Prompt (Hertz et al., 2022) manipulates cross-attention maps for precise control; MasaCtrl (Cao et al., 2023) adjusts self-attention to ensure consistency; and Direct Inversion (Ju et al., 2024) separates source and target diffusion to better preserve content. However, these approaches often require extra inference steps or feature extraction, increasing computational cost.

Beyond inversion-based methods, recent works explore the inversion-free editing, avoiding explicit reconstruction of the latent noise. InfEdit (Xu et al., 2024) introduces an inversion-free editing framework that reconstructs a consistent denoising path from noise, enabling fast and semantically coherent edits. FlowChef (Patel et al., 2024) exploits the straightness of RF models trajectories to propose a gradient-free, inversion-free steering strategy, significantly reducing computational costs for editing. FlowEdit (Kulikov et al., 2025) introduces a direct ODE mapping between source and target distributions, enabling efficient editing. These approaches substantially improve efficiency, but their reliance on approximate trajectory manipulation can lead to reduced fine-grained consistency. In contrast, `mimic-CFG` preserves the controllability of inversion-based pipelines while requiring no auxiliary models or structured inputs. By reducing inversion error through `PMI`, it enables accurate semantic edits with fewer sampling steps, striking a practical balance between efficiency and fine-grained consistency.

## 3 PRELIMINARIES

Before introducing our method, we present preliminaries on RF model training and sampling, the inversion process, and the inherent instability of existing inversion methods.

### 3.1 RECTIFIED FLOW

Rectified Flow (RF (Liu et al., 2022) is built on ODEs and learns an almost constant velocity field over the trajectory, enabling faster and more stable sampling. It maps real data $\mathbf{z}_0 \sim p_{\text{data}}$ to Gaussian noise $\mathbf{z}_1 \sim \mathcal{N}(0, \mathbf{I})$ via the following ODE:

$$\frac{d\mathbf{z}_t}{dt} = v_{\boldsymbol{\theta}}(\mathbf{z}_t, t), \quad t \in [0, 1]. \tag{1}$$

Here, $v_{\boldsymbol{\theta}}$ is a time-dependent velocity field parameterized by a neural network. The model is trained to match the ideal direction $\mathbf{z}_1 - \mathbf{z}_0$ throughout the trajectory by minimizing the following loss:

$$\mathcal{L}_{\text{RF}}(\boldsymbol{\theta}) = \mathbb{E}_{\mathbf{z}_0, \mathbf{z}_1, t} \left[ \|v_{\boldsymbol{\theta}}(\mathbf{z}_t, t) - (\mathbf{z}_1 - \mathbf{z}_0)\|_2^2 \right], \tag{2}$$

where $\mathbf{z}_t = (1 - t)\mathbf{z}_0 + t\mathbf{z}_1$ denotes a linear interpolation between $\mathbf{z}_0$ and $\mathbf{z}_1$. During inference, generation involves solving the learned ODE in reverse—from $\mathbf{z}_1$ back to $\mathbf{z}_0$. Using Euler's method,

the reverse update from $t_i$ to $t_{i-1}$ is given by:

$$\mathbf{z}_{t_{i-1}} = \mathbf{z}_{t_i} + (t_{i-1} - t_i)v_{\boldsymbol{\theta}}(\mathbf{z}_{t_i}, t_i), \tag{3}$$

where $\{t_i \in [0,1]\}_i$ is a decreasing sequence.

## 3.2 DDIM INVERSION

DDIM inversion (Euler scheduler) performs the reverse of the sampling process, mapping a real image into its latent noise. Based on the DDIM update rule, the inversion step of Euler scheduler using approximation is given by:

$$\mathbf{z}_{t_i} = \mathbf{z}_{t_{i-1}} + (\sigma_{t_i} - \sigma_{t_{i-1}})v_\theta(\mathbf{z}_{t_{i-1}}, t_{i-1}) \tag{4}$$

where $\sigma_{t_i}$ and $\sigma_{t_{i-1}}$ are scheduling parameters.

## 3.3 INSTABILITY IN INVERSION

(Zhang et al., 2025) show that using ODEs to map noise to images is inherently unstable in high dimensions: small perturbations in the latent space can cause large reconstruction errors. To quantify this effect, they introduce the geometric mean instability coefficient:

$$\bar{\mathcal{E}}_F(\mathbf{z}) = \left( \prod_{i=1}^{n} \frac{|J_F(\mathbf{z}_1)\mathbf{u}_i|}{|\mathbf{u}_i|} \right)^{\frac{1}{n}}, \tag{5}$$

where $J_F(\mathbf{z})$ is the Jacobian of $\mathbf{z}$ and $\mathbf{u}_i$ are orthonormal vectors, and when $\bar{\mathcal{E}}_F(\mathbf{z}) > 1$, it indicates that the mapping amplifies any infinitesimal perturbation. They prove that for any threshold $M > 1$, instability occurs with probability:

$$\mathcal{P}_M := \pi_{\text{real}} \left( \left\{ \mathbf{z} : \bar{\mathcal{E}}_F(F^{-1}(\mathbf{z})) > M \right\} \right) \geq 1 - \epsilon - \delta, \tag{6}$$

where

$$\epsilon := \pi_{\text{real}} \left( \left\{ \mathbf{z} : p_{\text{gen}}(\mathbf{z}) \geq \frac{1}{(2\pi M^2)^{n/2}} e^{-\frac{2n+3\sqrt{2n}}{2}} \right\} \right),$$
$$\delta := \pi_{\text{real}} \left( \left\{ \mathbf{z} : \|F^{-1}(\mathbf{z})\|^2 > 2n + 3\sqrt{2n} \right\} \right). \tag{7}$$

Here, $\pi_{\text{real}}$ is the real data distribution, and as dimension $n$ increases, both $\epsilon$ and $\delta$ vanish, implying that instability is almost guaranteed in high-dimensional ODE mappings.

## 3.4 INVERSION EVALUATION WITHOUT PROMPT BIAS

The inversion quality is ultimately assessed through downstream tasks such as reconstruction or editing. These downstream processes typically rely on text prompts or other conditional inputs. As a result, the observed reconstruction or editing performance may be influenced not only by the inversion accuracy but also by the quality and relevance of the conditioning prompt (Dong et al., 2023b; Li et al., 2025b). This makes conditional evaluation insufficient for isolating inversion errors, since prompt mismatch or weak textual alignment can obscure the true stability of the inversion trajectory.

To avoid the confounding effect introduced by prompts and to obtain a more faithful measure of inversion quality, we additionally report reconstruction performance under an unconditional setting. This prompt-free evaluation isolates the inversion process and reveals trajectory drift or accumulated errors that may be masked in conditional reconstruction. This diagnostic perspective offers a clearer assessment of inversion stability and complements the conditional metrics used in downstream editing tasks.

## 4 METHODOLOGY

### 4.1 PROXIMAL-MEAN INVERSION

As detailed in Sec. 3.3, RF in inversion sampling is susceptible to errors arising from intrinsic instability, often leading to suboptimal results. We posit that these errors stem from the disruption of the

originally near-constant velocity field during inversion, which compromises the stability essential for both inversion and reconstruction.

---

**Algorithm 1** Proximal-Mean Inversion (PMI): Euler Case

**Input**: velocity function $v_{\boldsymbol{\theta}}(\cdot)$, image $\mathbf{z}_0$, time schedule $\mathcal{T} = \{t_0 = 0, \ldots, t_N = 1\}$.

**Output**: $\hat{\mathbf{z}}_1$

1: $\mathbf{x}_{\text{dist}} = \mathbf{0}$
2: **for** $i = 0$ to $N - 1$ **do**
3: $\quad \mathbf{v}_{t_i} = v_{\theta}(\mathbf{z}_{t_i}, t_i)$
4: $\quad \mathbf{x}_{\text{dist}} += (t_{i+1} - t_i)\mathbf{v}_{t_i}$
5: $\quad \bar{\mathbf{v}}_{t_i} = \dfrac{1}{t_{i+1} - t_0}\mathbf{x}_{\text{dist}}$
6: $\quad \hat{\mathbf{v}}_{t_i} = \mathbf{v}_{t_i} - r_{t_i}\dfrac{\nabla F(\mathbf{v}_{t_i})}{\|\nabla F(\mathbf{v}_{t_i})\|_2}$
7: $\quad \hat{\mathbf{z}}_{t_{i+1}} = \hat{\mathbf{z}}_{t_i} + (t_{i+1} - t_i)\hat{\mathbf{v}}_{t_i}$
8: **end for**
9: **return** $\hat{\mathbf{z}}_1$

---

**Algorithm 2** `mimic-CFG` Editing: Euler Case

**Input**: velocity function $v_{\boldsymbol{\theta}}(\cdot)$, image $\mathbf{z}_0$, time schedule $\mathcal{T} = \{t_0 = 0, \ldots, t_N = 1\}$, interpolation weight $w$.

**Output**: $\hat{\mathbf{z}}_0$

1: $\hat{\mathbf{z}}_1 = \texttt{PMI}(v_{\theta}, \mathbf{z}_0, t)$
2: $\mathbf{x}_{\text{dist}} = \mathbf{0}$
3: **for** $i = N$ to $1$ **do**
4: $\quad \mathbf{v}_{t_i} = v_{\theta}(\hat{\mathbf{z}}_{t_i}, t_i)$
5: $\quad \mathbf{x}_{\text{dist}} += (t_{i-1} - t_i)\mathbf{v}_{t_i}$
6: $\quad \bar{\mathbf{v}}_{t_i}^{\text{edit}} = \dfrac{1}{t_{i-1} - t_N}\mathbf{x}_{\text{dist}}$
7: $\quad \bar{\mathbf{v}}_{t_i}^{\text{proj}} = \dfrac{\mathbf{v}_{t_i}^{\top}\bar{\mathbf{v}}_{t_i}^{\text{edit}}}{\|\bar{\mathbf{v}}_{t_i}^{\text{edit}}\|_2^2}\bar{\mathbf{v}}_{t_i}^{\text{edit}}$
8: $\quad \hat{\mathbf{v}}_{t_i} = (1 - w)\bar{\mathbf{v}}_{t_i}^{\text{proj}} + w\,\mathbf{v}_{t_i}$
9: $\quad \hat{\mathbf{z}}_{t_{i-1}} = \hat{\mathbf{z}}_{t_i} + (t_{i-1} - t_i)\hat{\mathbf{v}}_{t_i}$
10: **end for**
11: **return** $\hat{\mathbf{z}}_0$

---

To address this issue, we gradually correct the disturbed velocity field towards a more stable and near-constant one. Following the principle of global consistency, we guide velocity correction using an average velocity as a reference. Let $\mathbf{v}_{t_i} = v_{\boldsymbol{\theta}}(\mathbf{z}_{t_i}, t_i)$ denote the predicted velocity at time $t_i$, the average velocity from the initial time $t_0$ to the current time $t_k$ is defined as:

$$\bar{\mathbf{v}}_{t_k} := \frac{1}{t_{k+1} - t_0}\sum_{i=0}^{k}(t_{i+1} - t_i)\mathbf{v}_{t_i}, \tag{8}$$

for $k \in \{0, \ldots, N - 1\}$. To stabilize the perturbed velocity field, we employ a proximal objective using the average velocity:

$$F(\mathbf{v}) = \|\mathbf{v} - \mathbf{v}_{t_{k-1}}\|_1 + \frac{1}{2\lambda}\|\mathbf{v} - \bar{\mathbf{v}}_{t_k}\|_2^2. \tag{9}$$

The corrected velocity $\hat{\mathbf{v}}_{t_k}$ at timestep $t_k$ is then

$$\hat{\mathbf{v}}_{t_k} = \text{prox}_{\lambda}(\bar{\mathbf{v}}_{t_k}) = \arg\min_{\mathbf{v}} F(\mathbf{v}), \tag{10}$$

equation 10 balances two objectives with a trade-off factor $\lambda$ (results see Appendix B.1 and B.2):

- $\|\mathbf{v} - \mathbf{v}_{t_{k-1}}\|_1$: encourages local consistency by keeping the corrected velocity close to the previous step. An analysis experiment on the norm choice in $\|\mathbf{v} - \mathbf{v}_{t_{k-1}}\|_p$ (see Sec. 5.4) shows that $p = 1$ yields the best performance.

- $\|\mathbf{v} - \bar{\mathbf{v}}_{t_k}\|_2^2$: encourages the velocity toward the global average direction $\bar{\mathbf{v}}_{t_k}$.

However, solving equation 10 exactly at each timestep is computationally expensive. To improve efficiency, we approximate the objective with a first-order Taylor expansion around the current velocity $\mathbf{v}_{t_k}$:

$$F(\mathbf{v}) \approx F(\mathbf{v}_{t_k}) + \nabla F(\mathbf{v}_{t_k})^{\top}(\mathbf{v} - \mathbf{v}_{t_k}). \tag{11}$$

As a result, the constrained optimization problem becomes:

$$\arg\min_{\mathbf{v}}\nabla F(\mathbf{v}_{t_k})^{\top}(\mathbf{v} - \mathbf{v}_{t_k})$$
$$\text{s.t. } \|\mathbf{v} - \mathbf{v}_{t_k}\|^2 = r_{t_k}^2. \tag{12}$$

Solving equation 12 yields a closed-form update (proof in Appendix A.1):

$$\hat{\mathbf{v}}_{t_k} = \mathbf{v}_{t_k} - r_{t_k}\frac{\nabla F(\mathbf{v}_{t_k})}{\|\nabla F(\mathbf{v}_{t_k})\|_2}. \tag{13}$$

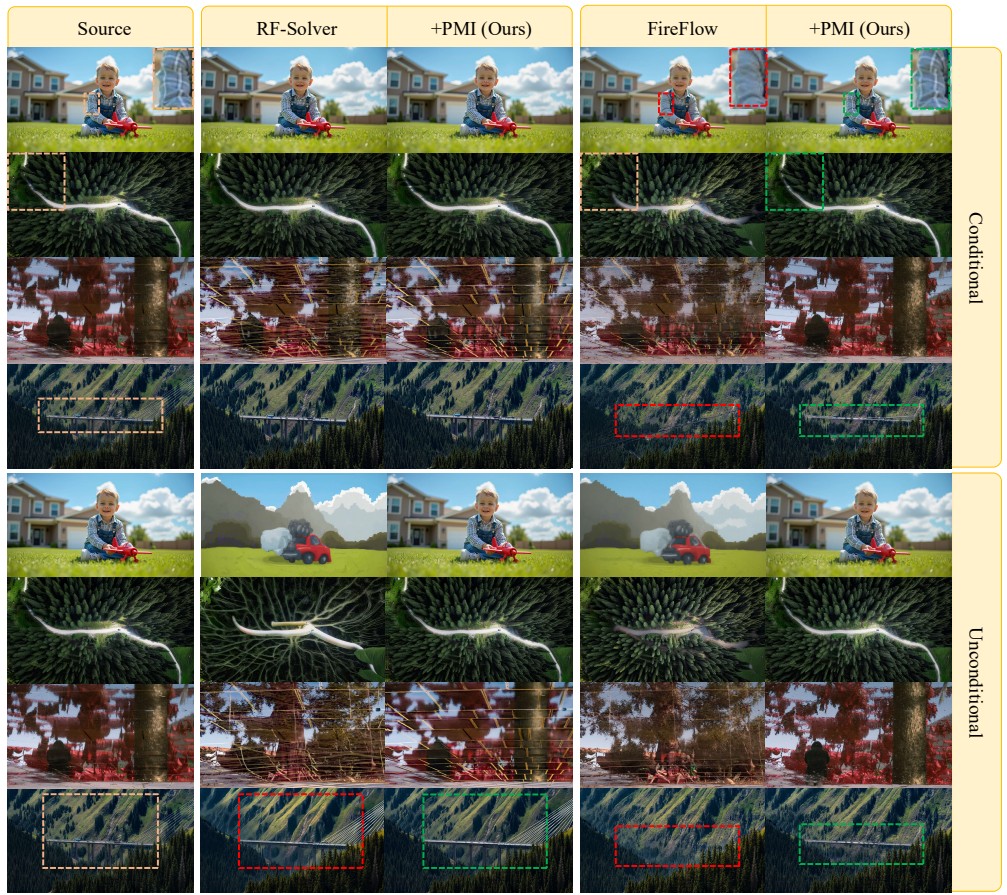

Figure 1: Qualitative comparison results for the inversion and reconstruction task on PIE-Bench. Baselines enhanced with our method consistently outperform their vanilla counterparts, particularly under the unconditional setting.

Here, the gradient direction is normalized, so the update magnitude depends only on the radius $r_{t_k}$. Moreover, we derive Proposition 1, which provides a condition on $r_{t_k}$ to ensure stability during inversion, based on probability bounds for low-density regions in instability theory.

**Proposition 1.** *(Stability Condition) Suppose the inverted latent vector $\hat{\mathbf{z}}_1 \in \mathbb{R}^n$ follows a Gaussian distribution*

$$\hat{\mathbf{z}}_1 \sim \mathcal{N}\big(0,\ \sigma^2 \mathbf{I}\big),$$

*and define $r = \sqrt{n}\,\sigma$, where $n$ is the latent dimension and $\sigma > 0$ is the scaling factor. The radius that can make the sampling points fall within the high-density region satisfies*

$$r_i = \sqrt{2n + 3\sqrt{2n}\frac{\Delta t_i}{T}} + \epsilon, \tag{14}$$

*where $T$ is the total time, $\Delta t_i = t_i - t_{i-1}$, $\epsilon > 0$ and $r_i$ denotes the step-wise correction radius at the $i$-th timestep.*

The full procedure is summarized in Algorithm 1 and referred to as `PMI`. Notably, `PMI` can be seamlessly integrated into existing inversion methods (*e.g.*, RF-Solver and FireFlow); here, we illustrate it using Euler's method. It requires no additional training and effectively stabilize inversion.

### 4.2 MIMIC-CFG EDITING

Although our instability theory is primarily developed for the Inversion and Reconstruction phases, we observe that similar instability can propagate into the Editing stage due to their shared reliance

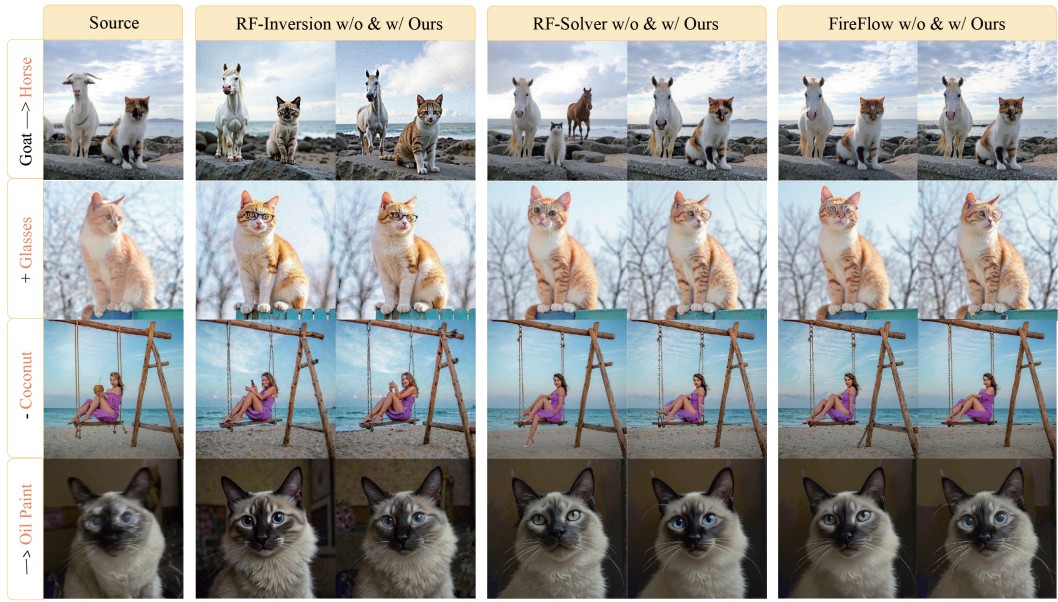

Figure 2: Qualitative comparison results for the editing task on PIE-Bench. The leftmost column shows the source images. Each of the three columns on the right contains two sets of results: the left side shows outputs from the vanilla baseline, while the right side shows outputs enhanced with our PMI and mimic-CFG methods. The results show that our methods can enhance the background preservation and editing quality in different editing categories.

on the velocity field. Therefore, we apply our correction strategy during Editing as well, to mitigate residual errors.

Similarly, for the editing stage (traversing from $t_N = 1$ to $t_0 = 0$), we define the average velocity up to timestep $t_k$ as:

$$\bar{\mathbf{v}}_{t_k}^{\text{edit}} := \frac{1}{t_{k-1} - t_N} \sum_{i=N}^{k} (t_{i-1} - t_i) v_\theta(\mathbf{z}_{t_i}, t_i). \tag{15}$$

At each timestep $t_k$, we shift the predicted velocity $\mathbf{v}_{t_k}$ toward the average velocity $\bar{\mathbf{v}}_{t_k}^{\text{edit}}$ to enhance stability. We aim to achieve this through interpolation. Prior study (Fan et al., 2025) has shown that interpolating between $\mathbf{v}_{t_k}$ and its projection onto $\bar{\mathbf{v}}_{t_k}^{\text{edit}}$ performs better than direct interpolation between the two vectors. We also empirically verify this finding in our setting (see Appendix B.8). Therefore, we apply the shortest path update along this direction: we first compute the projection of $\mathbf{v}_{t_k}$ onto the direction of $\bar{\mathbf{v}}_{t_k}^{\text{edit}}$, as follows:

$$\bar{\mathbf{v}}_{t_k}^{\text{proj}} = \frac{\mathbf{v}_{t_k}^\top \bar{\mathbf{v}}_{t_k}^{\text{edit}}}{\|\bar{\mathbf{v}}_{t_k}^{\text{edit}}\|_2^2} \bar{\mathbf{v}}_{t_k}^{\text{edit}}. \tag{16}$$

This projected vector $\bar{\mathbf{v}}_{t_k}^{\text{proj}}$ represents the shortest correction path from $v_{t_k}$ towards $\bar{\mathbf{v}}_{t_k}^{\text{edit}}$. Subsequently, we perform linear interpolation between $\bar{\mathbf{v}}_{t_k}^{\text{proj}}$ and $\mathbf{v}_{t_k}$ to obtain the corrected velocity:

$$\hat{\mathbf{v}}_{t_k} = (1 - w)\bar{\mathbf{v}}_{t_k}^{\text{proj}} + w \cdot \mathbf{v}_{t_k}. \tag{17}$$

Here, $w \in [0, 1]$ is an interpolation parameter used to control the degree of correction. An analysis experiment of $w$ (see Sec. B.1) shows that $w = 0.94$ yields the best performance. It is worth noting that if we treat $\bar{\mathbf{v}}_{t_k}^{\text{proj}}$ as the "unconditional" velocity and $\mathbf{v}_{t_k}$ as the "conditional" velocity, the above interpolation form closely resembles Classifier-Free Guidance (CFG) (Ho & Salimans, 2022). Therefore, we refer to this method as `mimic-CFG`.

We present the detailed algorithmic procedure in Algorithm 2. Starting from a corrected initial state obtained via `PMI`, we perform a reverse Euler process over the editing timesteps. The method introduces no additional training and improves stability during editing by aligning the velocity field with the average direction.

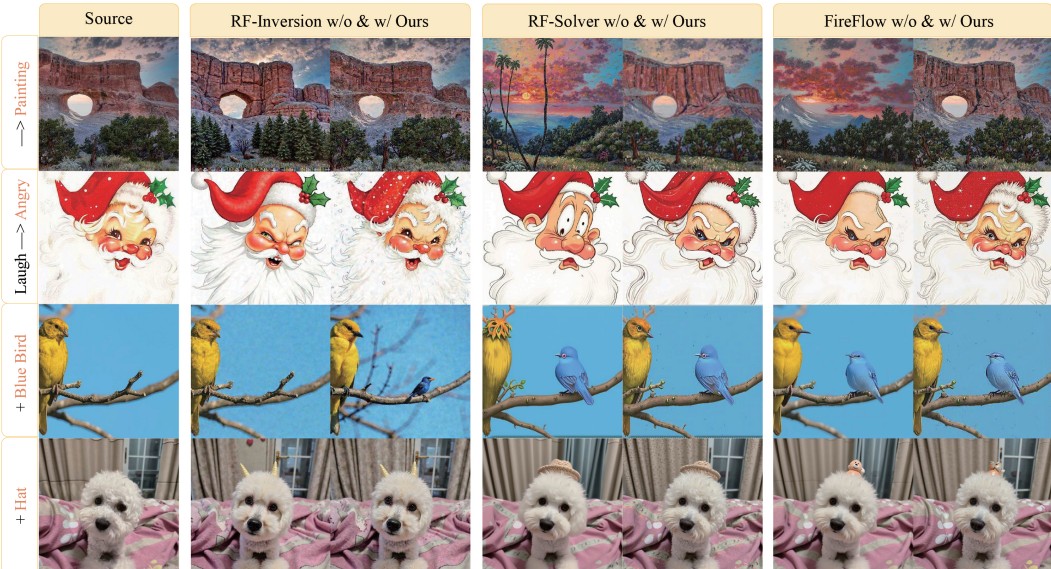

Figure 3: Qualitative comparison of editing results on PIE-Bench (first three) and real-world (last) images. Our method achieves superior adaptation in both synthetic and real-world editing scenarios.

Table 1: Quantitative results for inversion and reconstruction task. Our method consistently outperforms baseline models, achieving higher PSNR and SSIM, and lower MSE and LPIPS. Best results are in bold.

| Method | Conditional | | | | Unconditional | | | |
|---|---|---|---|---|---|---|---|---|
| | PSNR ↑ | $\text{LPIPS}^{\downarrow}_{\times 10^3}$ | $\text{MSE}^{\downarrow}_{\times 10^4}$ | $\text{SSIM}^{\uparrow}_{\times 10^2}$ | PSNR ↑ | $\text{LPIPS}^{\downarrow}_{\times 10^3}$ | $\text{MSE}^{\downarrow}_{\times 10^4}$ | $\text{SSIM}^{\uparrow}_{\times 10^2}$ |
| Euler | 22.10 | **168.48** | 96.58 | 79.36 | 21.96 | 189.26 | 101.57 | 77.45 |
| + PMI (ours) | **22.56** | 179.10 | **76.17** | **79.39** | **23.26** | **166.71** | **83.59** | **79.92** |
| Heun | 29.16 | 63.62 | 30.76 | 89.93 | 27.76 | 78.84 | 40.42 | 88.51 |
| + PMI (ours) | **30.38** | **53.25** | **21.51** | **90.34** | **29.86** | **62.67** | **32.44** | **90.26** |
| RF-Solver | 29.17 | 63.63 | 28.18 | 90.01 | 27.81 | 78.46 | 39.68 | 88.53 |
| + PMI (ours) | **30.72** | **53.69** | **22.42** | **91.23** | **29.87** | **62.66** | **29.88** | **90.29** |
| FireFlow | 29.72 | 55.87 | 23.05 | 90.86 | 28.87 | 66.04 | 27.03 | 89.76 |
| + PMI (ours) | **30.42** | **53.48** | **18.44** | **91.22** | **29.73** | **62.51** | **23.72** | **90.34** |

## 5 EXPERIMENTS

### 5.1 SETUP

**Baseline.** To demonstrate the effectiveness of our method, we selected several fundamental and commonly used RF-based baselines. For the inversion and reconstruction tasks, we incorporate our method into four solvers, Euler, Heun, RF-Solver (Wang et al., 2025a), and FireFlow (Deng et al., 2024), and compare their performance against their respective vanilla counterparts. For the editing task, we apply our method to the same four solvers as well as RF-Inversion (Rout et al., 2025), resulting in five edited variants, and evaluate them against their vanilla counterparts.

**Implementation details.** In all experiments, we use Flux.1-dev (Labs et al., 2025) as the underlying model for both reconstruction and editing tasks. For inversion and reconstruction experiments, we ensure that each baseline method, both with and without our proposed velocity correction strategy (PMI), shares the same NFE within each group to ensure fair comparison. For the editing task, we adopt the best settings reported in FireFlow for RF-Inversion, RF-Solver, and FireFlow, while maintaining the NFE around 50 for both the Euler and Heun samplers. More details see Appendix C.

**Evaluate metrics.** All experiments are conducted on the PIE-Bench (Ju et al., 2024), which consists of 700 images spanning 10 different editing categories. For the inversion and reconstruction task, we evaluate reconstruction quality using four widely adopted metrics: Peak Signal-to-

Table 2: Quantitative results for editing task. Our methods consistently improve CLIP similarity while maintain and slightly improve the background preservation metrics PSNR and SSIM. Best results are in bold and identical results are underlined.

| Method | Structure Distance$^{\downarrow}_{\times 10^3}$ | Background Preservation | | | | CLIP Similarity | | NFE $\downarrow$ |
| | | PSNR $\uparrow$ | LPIPS$^{\downarrow}_{\times 10^3}$ | MSE$^{\downarrow}_{\times 10^4}$ | SSIM$^{\uparrow}_{\times 10^2}$ | Whole $\uparrow$ | Edited $\uparrow$ | |
| Euler | 50.08 | 20.57 | 187.47 | 135.03 | 76.02 | 25.73 | 23.00 | 50 |
| + Ours | **46.06** | **21.41** | **185.55** | **121.78** | **76.04** | **25.87** | **23.01** | 50 |
| Heun | 44.51 | 20.51 | 176.09 | 137.53 | 77.45 | 25.79 | 23.08 | 48 |
| + Ours | **42.24** | **20.56** | **175.27** | **124.57** | **77.57** | **25.91** | **23.09** | 48 |
| RF-Inversion | 63.41 | 18.03 | 252.89 | 21.44 | 62.65 | 25.02 | 22.56 | 56 |
| + Ours | **63.39** | **18.04** | **247.13** | 21.44 | **62.94** | **25.44** | **22.83** | 56 |
| RF-Solver | 33.71 | 21.29 | 156.51 | 96.24 | 79.63 | 25.29 | 22.61 | 60 |
| + Ours | **32.69** | **22.18** | **146.85** | **85.64** | **80.39** | **25.49** | **22.72** | 60 |
| FireFlow | 29.39 | 22.40 | 136.49 | 79.35 | 80.78 | 25.35 | 22.52 | 18 |
| + Ours | **28.01** | **22.79** | **128.55** | **78.02** | **81.76** | **25.50** | **22.68** | 18 |

Noise Ratio (PSNR) (Huynh-Thu & Ghanbari, 2008), Learned Perceptual Image Patch Similarity (LPIPS) (Zhang et al., 2018), Mean Squared Error (MSE), and Structural Similarity Index Measure (SSIM) (Wang et al., 2004). For the editing task, the same four metrics are used to assess background preservation in unedited areas. To evaluate editing fidelity, we introduce the CLIP similarity (Radford et al., 2021) for both the whole image and the edited regions. Furthermore, we employ Structure Distance (Ju et al., 2024) to evaluate edit-irrelative context preservation.

## 5.2 INVERSION AND RECONSTRUCTION

**Quantitative comparison.**

To prove the effectiveness of our proposed correction method, we compare it with various baseline models on inversion and reconstruction tasks, as shown in Tab. 1. The results indicate consistent improvements with our correction method (PMI) across all baseline samplers. Compared to their vanilla counterparts, our method achieves higher PSNR and SSIM, and lower MSE and LPIPS, with particularly notable improvements on Heun (PSNR improved by 1.22) and RF-Solver (PSNR improved by 1.55). Additionally, we evaluate the unconditional setting for each group, where our method maintains strong performance even in the absence of prompts. These results demonstrate the effectiveness of our approach in stabilizing the inversion process, enhancing reconstruction quality, and reducing the likelihood of falling into low-density regions. Additionally, we present the reconstruction error in Appendix B.10.

**Qualitative comparison.**

Fig. 1 demonstrates a qualitative comparison of the inversion and reconstruction task on RF-Solver and FireFlow, both with and without our method. Our approach significantly improves reconstruction quality in both conditional and unconditional settings. In the unconditional case, baseline models without our method exhibit noticeable degradation in detail and structural fidelity.

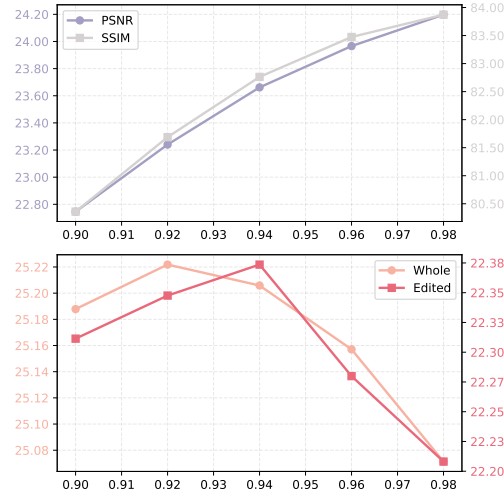

Figure 4: Analysis experiment of the parameter $w$ in mimic-CFG. PSNR and SSIM improve with moderate correction, while over correction (small $w$) harms both background preservation and editing quality.

## 5.3 Editing

**Quantitative comparison.** To validate the effectiveness of our method, we demonstrate the quantitative results for text-driven image editing on PIE-Bench in Tab. 2. Integrating our method with all baseline samplers results in clear improvements in background preservation metrics such as PSNR and SSIM, while maintaining a competitive CLIP similarity. For instance, our method significantly improves both CLIP similarity on RF-Inversion while maintaining and slightly enhancing background preservation. We also compared the same baselines with and without our method across different NFE. The results show that the baseline with our method achieves similar performance with significantly lower NFE (see in Appendix B.9). These results indicate that our method effectively corrects velocity, balancing background preservation and editing quality, while significantly reducing the number of sampling steps.

**Qualitative comparison.**

In Fig. 2, we compare three methods with and without our methods. We can see that the baseline methods often have poor background preservation or low editing quality. These problems usually need more sampling steps to improve. In contrast, our editing method fixes these issues while using same or fewer sampling steps.

Here, to further demonstrate the practical effectiveness of our method in real-world settings, we present additional qualitative comparative results on editing tasks, as shown in Fig. 3. The first three images are from PIE-Bench, while the last one is a real-world photo. The results demonstrate that our method better facilitates the model's adaptation to both benchmark and real-world image editing.

## 5.4 Analysis Experiments

**Analysis of $w$.** In Fig.4, we demonstrate the result of the analysis experiment of the parameter $w$ in our proposed `mimic-CFG` method. The result indicates that $w = 0.94$ provides the best balance between background preservation and editing quality. As $w$ increases, both PSNR and SSIM improve. However, as the velocity correction effect weakens (when $w$ decreases), the editability worsens.

**Analysis on the choice of norm.**

Tab. 3 shows the analysis experiment on the norm used in the $\|\mathbf{v} - \mathbf{v}_{t_{k-1}}\|_p$. We compare the setting with no such term, using the $L_1$ norm, and using the $L_2$ norm. The evaluation is conducted on PIE-Bench with PSNR, SSIM, and CLIP similarity for both the whole image and the edited region. The $L_1$ norm achieves slightly better performance on PSNR, SSIM and CLIP similarity for the edited region, indicating it provides a more robust regularization while maintaining local consistency.

Table 3: Analysis Experiment on the choice of norm. The $L_1$ norm achieves slightly better performance. Best results are in bold and identical results are underlined.

| Method | PSNR ↑ | SSIM$^{\uparrow}_{\times 10^2}$ | Whole ↑ | Edited ↑ |
|---|---|---|---|---|
| w/o norm | 23.66 | 82.76 | 25.20 | 22.37 |
| $L_1$-norm | **23.67** | **82.77** | 25.20 | **22.38** |
| $L_2$-norm | 23.65 | 82.75 | 25.18 | 22.32 |

## 6 Conclusion

In this paper, we introduce Proximal-Mean Inversion (`PMI`), a velocity correction method for RF-based models that mitigates disturbances in the velocity field and reduces accumulated errors during inversion. Furthermore, to prevent the corrected trajectory from deviating into low-density regions, we constrain each correction step within a spherical Gaussian, with the radius derived by us based on instability analysis. Building on this correction concept, we also present `mimic-CFG`, a lightweight correction strategy for editing tasks, which guides the predicted velocity towards the running average direction to achieve an improved balance between background preservation and editing fidelity. Both methods are training-free and plug-and-play. Extensive experiments on PIE-Bench demonstrate that our approaches significantly enhance reconstruction and editing quality, reduce sampling steps compared to state-of-the-art RF-based inversion methods.

ACKNOWLEDGEMENT

This work was supported by the National Natural Science Foundation of China (No. 6250070674) and the Zhejiang Leading Innovative and Entrepreneur Team Introduction Program (2024R01007).

ETHICS STATEMENT

**Data and Privacy.** All experiments in this paper are conducted on publicly available datasets, including PIE-Bench, which consists of benchmark images for inversion and editing evaluation. These datasets contain no personally identifiable information (PII), sensitive user data, or proprietary content. We strictly follow the licenses of all datasets and models employed, ensuring that our usage complies with both legal and ethical norms.

**Research Purpose and Potential Risks.** Our proposed methods, Proximal-Mean Inversion (`PMI`) and `mimic-CFG`, are designed to improve the stability and controllability of rectified-flow (RF)-based generative models during inversion and editing. The contributions are methodological and theoretical, aiming to reduce approximation errors and enhance reconstruction fidelity and editing quality. Nevertheless, generative models can be misused in ways that cause harm, such as generating disinformation, creating manipulated media without consent, or reinforcing existing biases in training data. We acknowledge these risks and emphasize that our work is intended for academic research and responsible downstream applications only. We explicitly discourage malicious use of the proposed methods, particularly in domains where manipulation may infringe on human rights, amplify stereotypes, or spread harmful misinformation.

**Fairness and Bias.** Since our work builds upon pre-trained generative models, it may inherit dataset biases or artifacts embedded in these models. Our methods do not introduce additional bias mitigation, but by improving inversion and editing stability, they may reduce certain error-induced distortions in downstream tasks. We encourage further studies to combine our correction strategies with fairness-aware training or evaluation pipelines to ensure more equitable outcomes.

**Environmental and Computational Impact.** All experiments were performed on a single NVIDIA A100 GPU with 40GB memory. Compared to training large generative models from scratch, our approach is training-free and thus imposes a relatively small computational and environmental footprint. We further optimized our experiments to minimize unnecessary runs and repeated evaluations. Nevertheless, we recognize the broader sustainability concerns of large-scale machine learning and encourage future research to adopt resource-conscious practices.

**Contribution to Society.** We believe that this research advances fundamental understanding of inversion and editing in generative models, offering new tools for trustworthy and efficient applications. Potential positive applications include image restoration, creative content generation with proper consent, and controllable editing that preserves structural fidelity. We expect these contributions to support the responsible development of AI systems that are transparent, reproducible, and beneficial to society.

**Conflicts of Interest.** The authors declare no conflicts of interest.

REPRODUCIBILITY STATEMENT

We have taken multiple steps to ensure the reproducibility of our work. First, we provide complete algorithmic descriptions of both Proximal-Mean Inversion (`PMI`) and `mimic-CFG` in Sec. 4. Additional theoretical details, such as proofs of stability conditions and error analysis, are included in Appendix A. Second, we report all experimental settings, including dataset specifications, evaluation metrics, and hyperparameter ranges, in Sec. 5 and Appendix C, with detailed analysis experiments provided in Appendix B. The computing infrastructure, including GPU, CPU, memory, and software environment, is documented in Appendix C.1 Finally, to further facilitate reproducibility, we provide anonymized source code as supplementary material, which implement our methods reported in the paper.

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

# A  MISSING PROOFS

## A.1  CLOSE FORM SOLUTION OF EQUATION 12

We first restate the equation as follows:

$$\underset{\mathbf{v}_{t_k}}{\operatorname{argmin}} \nabla F(\mathbf{v}_{t_k})(\mathbf{v} - \mathbf{v}_{t_k}) \\ \text{s.t.} \|\mathbf{v} - \mathbf{v}_{t_k}\|^2 = r^2 , \tag{A.1}$$

where $\mathbf{v}$ and $\mathbf{v}_{t_k}$ denote the corrected and current velocities at time $t_k$, respectively, and $r$ is the radius of the spherical Gaussian centered at $\mathbf{v}_{t_k}$. The optimal solution can be derived as follows.

First we reparameterize the constrain condition

$$\mathbf{v} = \mathbf{v}_{t_k} + r\mathbf{d}, \tag{A.2}$$

where $\|\mathbf{d}\|_2^2 = 1$, and $\mathbf{d}$ is the direction from $\mathbf{v}_{t_k}$ to $\mathbf{v}$, substituting this into the original objective, we obtain:

$$\begin{aligned} &\underset{\mathbf{v}_{t_k}}{\operatorname{argmin}} \nabla F(\mathbf{v}_{t_k})(\mathbf{v} - \mathbf{v}_{t_k}) \\ =&\underset{\mathbf{v}_{t_k}}{\operatorname{argmin}} \nabla F(\mathbf{v}_{t_k})(\mathbf{v}_{t_k} + r\mathbf{d} - \mathbf{v}_{t_k}) \\ =&\underset{\|\mathbf{d}\|_2^2=1}{\operatorname{argmin}} \ r\nabla F(\mathbf{v}_{t_k})^T \mathbf{d}. \end{aligned} \tag{A.3}$$

We now construct the Lagrangian function:

$$\mathcal{L}(\mathbf{d}, \lambda) = r\nabla F(\mathbf{v}_{t_k})^T d + \lambda(\|\mathbf{d}\|_2^2 - 1) \tag{A.4}$$

then we have the KKT condition

$$\begin{cases} \nabla \mathcal{L} = r\nabla F(\mathbf{v}_{t_k}) + 2\lambda\mathbf{d} \\ \quad\quad \|\mathbf{d}\|_2^2 = 1 \\ \quad\quad\quad \lambda > 0 \end{cases} . \tag{A.5}$$

Setting $\nabla \mathcal{L} = 0$ yields:

$$\mathbf{d} = -\frac{r\nabla F(\mathbf{v}_{t_k})}{2\lambda}. \tag{A.6}$$

Substituting into the constraint $\|\mathbf{d}\|_2^2 = 1$ and solving for $\lambda$, we get:

$$\lambda = \frac{r\|\nabla F(\mathbf{v}_{t_k})\|}{2}. \tag{A.7}$$

Thus, the optimal solution for $\mathbf{d}$ is:

$$\mathbf{d} = -\frac{\nabla F(\mathbf{v}_{t_k})}{\|\nabla F(\mathbf{v}_{t_k})\|}. \tag{A.8}$$

Therefore, the closed-form solution of Equation 11 is:

$$\mathbf{v} = \mathbf{v}_{t_k} - r\frac{\nabla F(\mathbf{v}_{t_k})}{\|\nabla F(\mathbf{v}_{t_k})\|}, \tag{A.9}$$

which equals to:

$$\hat{\mathbf{v}}_{t_k} = \mathbf{v}_{t_k} - r\frac{\nabla F(\mathbf{v}_{t_k})}{\|\nabla F(\mathbf{v}_{t_k})\|}. \tag{A.10}$$

## A.2  THE RANGE OF $r$ DURING THE INVERSION.

Inspired by Zhang et al. (2025) and Yang et al. (2024), we want to reduce the probability of the model falling into the low density regions by using the spherical Gaussian. Next we give the proof of Proposition 2

**Proposition 1.** *Suppose the inverted latent vector $\hat{\mathbf{z}}_1 \in \mathbb{R}^n$ follows a Gaussian distribution*

$$\hat{\mathbf{z}}_1 \sim \mathcal{N}\left(0, \; \sigma^2 \mathbf{I}\right),$$

*and define*

$$r = \sqrt{n}\,\sigma,$$

*where $n$ is the latent dimension and $\sigma > 0$ is the scaling factor. According to the instability theorem, the radius that can make the sampling points fall within the high-density region satisfies*

$$r_i = \sqrt{2n + 3\sqrt{2n}}\frac{\Delta t_i}{T} + \epsilon,$$

*where $T$ is the total time, $\Delta t_i = t_i - t_{i-1}$, $\epsilon > 0$ and $r_i$ denotes the step-wise correction radius at the $i$-th time step.*

*Proof.* We first numerically calculate the variance $\hat{\sigma}$ of $\hat{\mathbf{z}}_1$

$$\hat{\sigma}^2 = \frac{1}{n}\sum_{i=0}^{n}(\bar{\mathbf{z}}_{\mathbf{i}} - \hat{\mu})^2 = \frac{1}{n}\|\bar{\mathbf{z}}_{\mathbf{1}}\|^2, \tag{A.11}$$

where $\bar{\mathbf{z}}_{\mathbf{1}}$ and $\bar{\mathbf{z}}_{\mathbf{i}}$ denote the flattened representations of $\hat{\mathbf{z}}_{\mathbf{1}}$ and $\hat{z}_i$, respectively. According to the theorem, we have

$$\hat{\sigma}^2 n \geq 2n + 3\sqrt{2n}. \tag{A.12}$$

Since $r = \sqrt{n}\,\sigma$, this can be equivalently written as

$$r^2 \geq 2n + 3\sqrt{2n}. \tag{A.13}$$

Therefore, we obtain

$$r \geq \sqrt{2n + 3\sqrt{2n}}. \tag{A.14}$$

Suppose that $\forall \varepsilon > 0$ and set $\delta = \dfrac{\varepsilon}{r} > 0$. When

$$0 < s_i = \frac{\Delta t_i}{T} < \delta, \tag{A.15}$$

we have

$$r s_i < \varepsilon. \tag{A.16}$$

Define the local radius

$$r_i = r s_i + \varepsilon. \tag{A.17}$$

Then $r_i - r s_i = \varepsilon > 0$, and explicitly

$$r_i = \sqrt{2n + 3\sqrt{2n}}\frac{\Delta t_i}{T} + \varepsilon. \tag{A.18}$$

Hence, for every discrete index $i$, the lower bound on the radius remains strictly larger than $r s_i$, while approaching $r s_i$ as $\varepsilon \to 0$.

$\square$

### A.3  ERROR ANALYSIS

**Assumption 1.** *The velocity function $v_\theta(\cdot, \tau)$ is $C_1$-Lipschitz for $\forall \tau$, i.e., given a $\tau$,*

$$\|v_\theta(\zeta_1, \tau) - v_\theta(\zeta_2, \tau)\| \leq C_1\|\zeta_1 - \zeta_2\|, \quad \forall \zeta_1, \zeta_2.$$

**Assumption 2.** *The velocity function $v_\theta(\zeta, \cdot)$ is $C_2$-Lipschitz for $\forall \zeta$, i.e., given a $\zeta$,*

$$\|v_\theta(\zeta, \tau_1) - v_\theta(\zeta, \tau_2)\| \leq C_2\|\tau_1 - \tau_2\| \quad \forall \tau_1, \tau_2.$$

**Proposition 2.** *Suppose that the velocity field $v_\theta$ is Lipschitz, and there is a constant $L$ s.t. $\|\mathbf{z}_{t_i} - \mathbf{z}_{t_{i-1}}\|_2 \leq L\|t_i - t_{i-1}\|$, $\forall t_i, t_{i-1} \in [0, 1]$. Then for any two consecutive steps $t_{i-1}$ and $t_i$, the local error of inversion and reconstruction using our method is $\mathcal{O}(\Delta t_i^2)$, achieves the same local truncation error as the standard Euler method, where $\Delta t_i = t_i - t_{i-1}$.*

*Proof.* We consider the recursive relationship of the one-step error term:

$$\begin{aligned}
\|\mathbf{z}_{t_i} - \hat{\mathbf{z}}_{t_i}\| &\leq \|\mathbf{z}_{t_{i-1}} - \hat{\mathbf{z}}_{t_{i-1}}\| \\
&\quad + (t_i - t_{i-1})\big\|v_\theta(\mathbf{z}_{t_i}, t_i) - v_\theta(\hat{\mathbf{z}}_{t_{i-1}}, t_{i-1})\big\| \\
&\quad + (t_i - t_{i-1})\|r\frac{\nabla F(\hat{\mathbf{v}}_{t_i})}{\|\nabla F(\hat{\mathbf{v}}_{t_i})\|}\|.
\end{aligned} \tag{A.19}$$

First, expand the velocity field differences:

$$\begin{aligned}
&\big\|v_\theta(\mathbf{z}_{t_i}, t_i) - v_\theta(\hat{\mathbf{z}}_{t_{i-1}}, t_{i-1})\big\| \\
&\leq \big\|v_\theta(\mathbf{z}_{t_i}, t_i) - v_\theta(\mathbf{z}_{t_i}, t_{i-1})\big\| \\
&\quad + \big\|v_\theta(\mathbf{z}_{t_i}, t_{i-1}) - v_\theta(\hat{\mathbf{z}}_{t_{i-1}}, t_{i-1})\big\| \\
&\quad + \|r\|.
\end{aligned} \tag{A.20}$$

From the Lipschitz condition Asm.1 and Asm.2, we obtain:

$$\begin{aligned}
\big\|v_\theta(\mathbf{z}_{t_i}, t_{i-1}) - v_\theta(\hat{\mathbf{z}}_{t_{i-1}}, t_{i-1})\big\| &\leq C_1\|\mathbf{z}_{t_i} - \hat{\mathbf{z}}_{t_{i-1}}\|, \\
\big\|v_\theta(\mathbf{z}_{t_i}, t_i) - v_\theta(\mathbf{z}_{t_i}, t_{i-1})\big\| &\leq C_2\|t_i - t_{i-1}\|,
\end{aligned} \tag{A.21}$$

such that

$$\begin{aligned}
&\big\|v_\theta(\mathbf{z}_{t_i}, t_i) - v_\theta(\hat{\mathbf{z}}_{t_{i-1}}, t_{i-1})\big\| \\
&\leq C_2\|t_i - t_{i-1}\| + C_1\|\mathbf{z}_{t_i} - \hat{\mathbf{z}}_{t_{i-1}}\| + \|r\| \\
&= C_2\Delta t_i + C_1\|\mathbf{z}_{t_i} - \hat{\mathbf{z}}_{t_{i-1}}\| + \|r\| \\
&\leq C_2\Delta t_i + C_1(\|\mathbf{z}_{t_i} - \mathbf{z}_{t_{i-1}}\| + \|\mathbf{z}_{t_{i-1}} - \hat{\mathbf{z}}_{t_{i-1}}\|) + \|r\| \\
&= C_2\Delta t_i + C_1\Delta t_i + C_1\|\mathbf{z}_{t_{i-1}} - \hat{\mathbf{z}}_{t_{i-1}}\| + \|r\|
\end{aligned} \tag{A.22}$$

We define that $\mathcal{E} = \|\mathbf{z}_{t_{i-1}} - \hat{\mathbf{z}}_{t_{i-1}}\|$, the local error of $\|\mathbf{z}_{t_i} - \hat{\mathbf{z}}_{t_i}\|$ as

$$\mathcal{E}^{\text{local}} = (t_i - t_{i-1})\big\|v_\theta(\mathbf{z}_{t_i}, t_i) - v_\theta(\hat{\mathbf{z}}_{t_{i-1}}, t_{i-1})\big\|, \tag{A.23}$$

and substitute them back into the original inequality:

$$\mathcal{E}^{\text{local}} \leq (C_1 + C_2)\Delta t_i^2 + C_1\mathcal{E}\Delta t_i + \|r\|\Delta t_i. \tag{A.24}$$

We can avoid the accumulated error $\mathcal{E}$, then we have the local error:

$$\mathcal{E}^{\text{local}} \leq (C_1 + C_2)\Delta t_i^2 + \|r\|\Delta t_i. \tag{A.25}$$

Now we substitute the $r$ in Prop.2 without $\epsilon$, we have

$$\begin{aligned}
\mathcal{E}^{\text{local}} &\leq (C_1 + C_2)\Delta t_i^2 + \|\sqrt{2n + 3\sqrt{2n}}\frac{\Delta t_i}{T}\|\Delta t_i \\
&= (C_1 + C_2 + \frac{\sqrt{2n + 3\sqrt{2n}}}{T})\Delta t_i^2 \\
&= \mathcal{O}(\Delta t_i^2).
\end{aligned} \tag{A.26}$$

$\square$

The Proposition 2 indicates that our method will not increase the error compared to the baseline.

## A.4 ANALYSIS ON THE EXPANSION ORDER OF $F(\mathbf{v})$

To better understand the effect of using higher-order approximations in PMI, we analyze the Taylor expansion of the proximal objective $F(\mathbf{v})$ around the current velocity $\mathbf{v}_{t_k}$.

**Second-order expansion.** The second-order Taylor expansion of $F(\mathbf{v})$ around $\mathbf{v}_{t_k}$ is given by:

$$F(\mathbf{v}) \approx F(\mathbf{v}_{t_k}) + \nabla F(\mathbf{v}_{t_k})^\top(\mathbf{v} - \mathbf{v}_{t_k}) + \frac{1}{2}(\mathbf{v} - \mathbf{v}_{t_k})^\top\nabla^2 F(\mathbf{v}_{t_k})(\mathbf{v} - \mathbf{v}_{t_k}). \tag{A.27}$$

Following the constrained formulation in Eq. A.1, we obtain:

$$\underset{\|\mathbf{d}\|_2^2=1}{\arg\min} \quad \nabla F(\mathbf{v}_{t_k})^\top(\mathbf{v} - \mathbf{v}_{t_k}) + \frac{1}{2}(\mathbf{v} - \mathbf{v}_{t_k})^\top \nabla^2 F(\mathbf{v}_{t_k})(\mathbf{v} - \mathbf{v}_{t_k}). \tag{A.28}$$

Since $F$ consists of an $\ell_1$ regularization term and a quadratic term, its Hessian satisfies

$$\nabla^2 F(\mathbf{v}_{t_k}) = \frac{1}{\lambda}\mathbf{I}, \tag{A.29}$$

where the $\ell_1$ term contributes zero curvature almost everywhere. Plugging this structure into Eq. A.28, we arrive at:

$$\underset{\|\mathbf{d}\|_2^2=1}{\arg\min} \ \nabla F(\mathbf{v}_{t_k})^\top(\mathbf{v} - \mathbf{v}_{t_k}) + \frac{1}{2\lambda}\|\mathbf{v} - \mathbf{v}_{t_k}\|_2^2. \tag{A.30}$$

Using the spherical constraint reparameterization

$$\mathbf{v} = \mathbf{v}_{t_k} + r\mathbf{d}, \qquad \|\mathbf{d}\|_2 = 1, \tag{A.31}$$

we obtain

$$\underset{\|\mathbf{d}\|_2^2=1}{\arg\min} \ r\,\nabla F(\mathbf{v}_{t_k})^\top \mathbf{d} + \frac{r^2}{2\lambda}. \tag{A.32}$$

Since the second term is constant w.r.t. $\mathbf{d}$, the optimization reduces to

$$\underset{\|\mathbf{d}\|_2^2=1}{\arg\min} \ r\,\nabla F(\mathbf{v}_{t_k})^\top \mathbf{d}, \tag{A.33}$$

which is identical to Eq. A.3. This shows that the optimal direction of the update remains unchanged even when second-order information is included. Thus, the `PMI` update rule under second-order expansion is equivalent to the first-order case.

**Higher-order expansions.** For $n \geq 3$, the Taylor expansion of $F$ contains higher-order derivatives:

$$\nabla^n F(\mathbf{v}_{t_k}). \tag{A.34}$$

However, $F(\mathbf{v})$ is composed of an $\ell_1$ term (piecewise linear) and a quadratic term (constant Hessian). Therefore,

$$\nabla^n F(\mathbf{v}_{t_k}) = \mathbf{0}, \qquad \forall n \geq 3. \tag{A.35}$$

This implies that all higher-order Taylor terms vanish identically. Consequently, third-order and higher-order expansions produce the same optimization objective as the second-order case, and thus yield exactly the same closed-form update direction.

## B EXPERIMENTS

### B.1 ANALYSIS EXPERIMENT OF $\lambda$ ON RECONSTRUCTION

To investigate the optimal choice of $\lambda$ for the inversion and reconstruction task, we conduct an analysis experiment under the conditional case. We aim to select a $\lambda$ value that yields higher PSNR and SSIM while achieving lower LPIPS and MSE. As shown in Fig. B.1, the choice of the trade-off factor $\lambda$ significantly influences the performance on the inversion and reconstruction task. As $\lambda$ increases, both PSNR and SSIM generally improve, with $\lambda = 10$ achieving the highest values. Meanwhile, LPIPS and MSE exhibit a decreasing trend, and $\lambda = 10$ yields the lowest values across both metrics. These results indicate that $\lambda = 10$ provides the best balance between preserving perceptual quality and ensuring pixel-level fidelity.

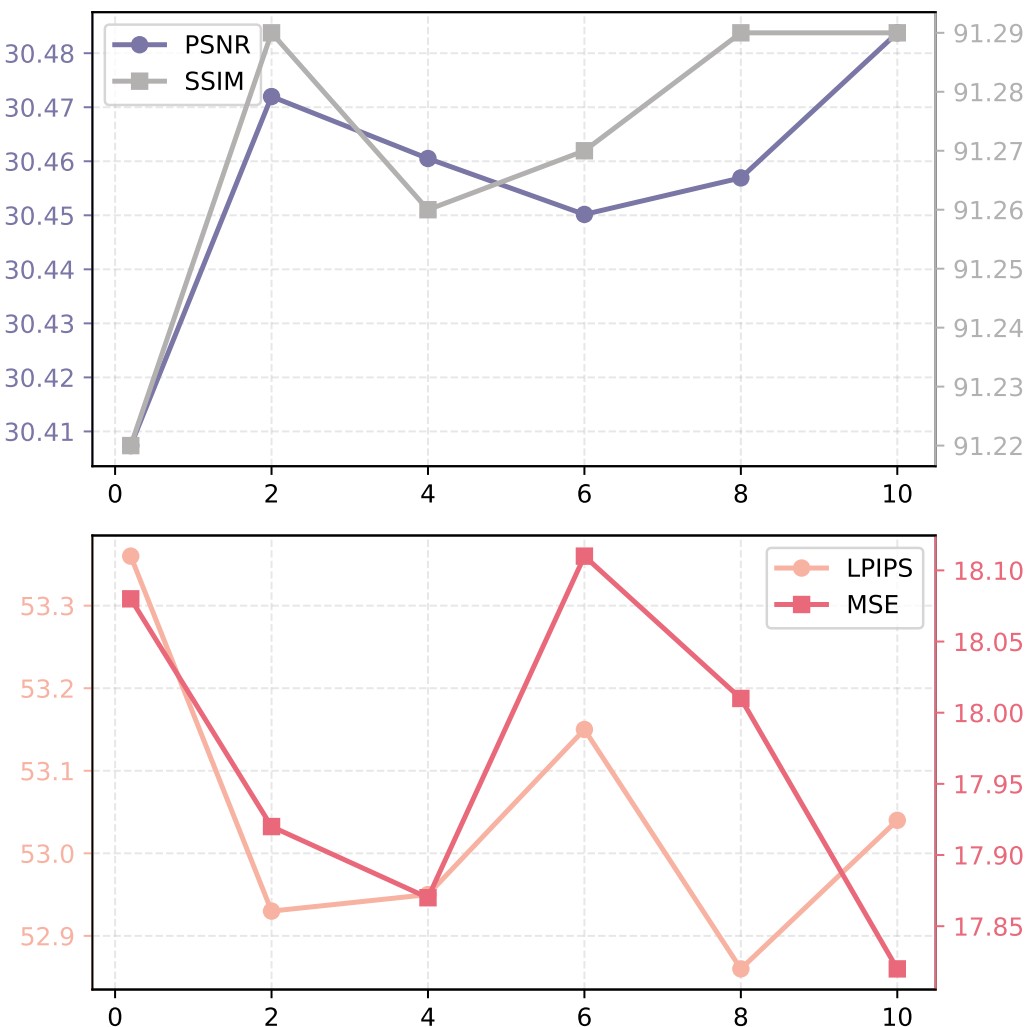

Figure B.1: Analysis experiment on the effect of the proximal operator parameter $\lambda$. Increasing $\lambda$ leads to improvements in PSNR and SSIM, while excessively large values result in overcorrection, compromising editing quality.

## B.2 ANALYSIS EXPERIMENT OF $\lambda$ ON EDITING

We study the effect of the hyperparameter $\lambda$ in the proximal operator for editing tasks. As illustrated in Fig. B.2, we vary $\lambda$ from 0 to 1 and report PSNR, SSIM, and CLIP similarity (Whole and Edited). As $\lambda$ increases, both PSNR and SSIM exhibit slight improvements, indicating that stronger proximal correction enhances image fidelity. However, the CLIP similarity results suggest that overly large $\lambda$ can degrade editing performance: Whole similarity drops gradually, and Edited similarity becomes less stable, reflecting weakened editability and overcorrection. These results demonstrate a trade-off between reconstruction accuracy and editing effectiveness. We observe that $\lambda = 1$ offers a good balance and adopt it as the default setting for editing tasks for FireFlow.

We also study the effect of the hyperparameter $\lambda$ on the performance of the `PMI` solver when used with other solvers, including Euler, Heun, and RF-Solver. As shown in Fig. B.3, Fig. B.4, and Fig. B.5, the optimal $\lambda$ for Euler, Heun, and RF-Solver are $\lambda = 8$, $\lambda = 1$, and $\lambda = 1$, respectively.

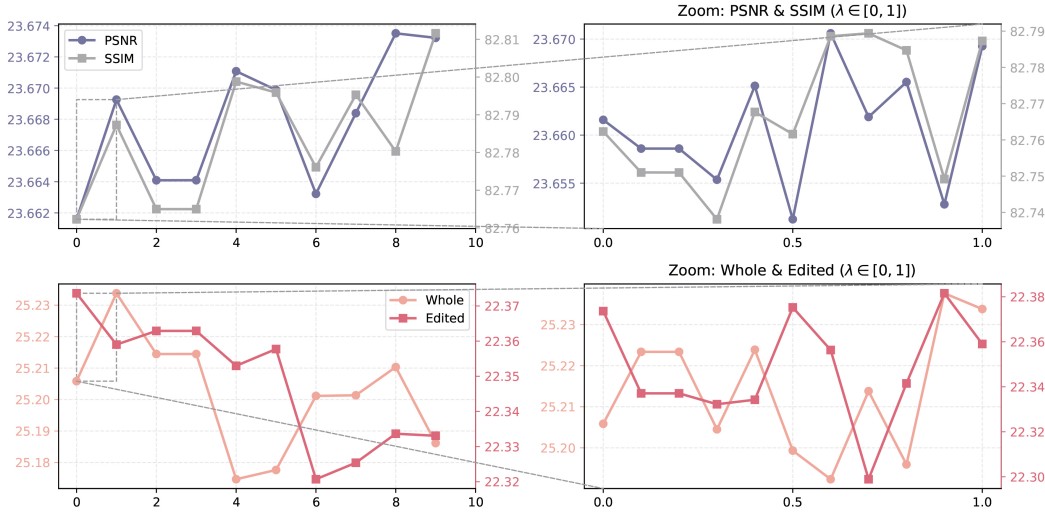

Figure B.2: Analysis experiment on the effect of the proximal operator parameter $\lambda$ with the solver **FireFlow**. Increasing $\lambda$ leads to improvements in PSNR and SSIM, while excessively large values result in overcorrection, compromising editing quality.

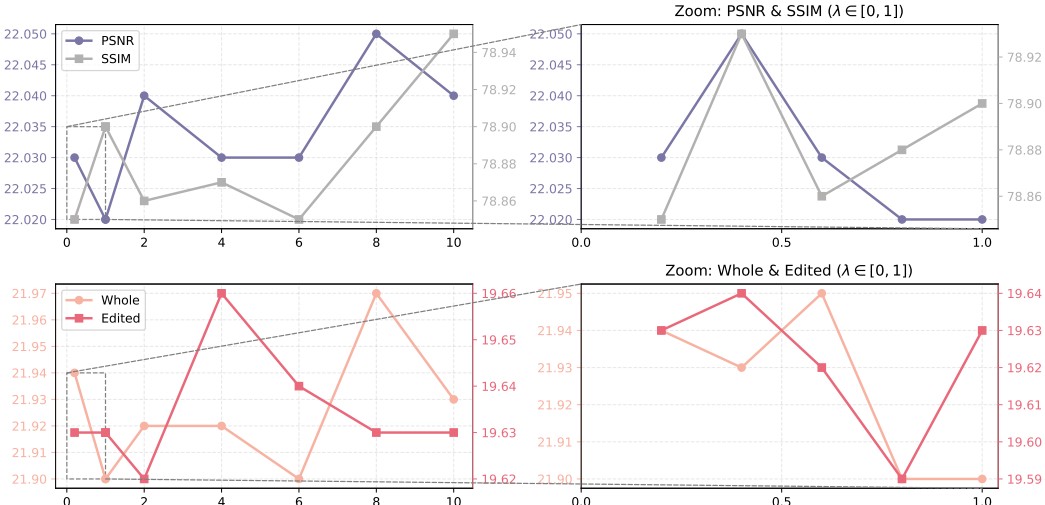

Figure B.3: Analysis experiment on the effect of the proximal operator parameter $\lambda$ with the solver **Euler**.

## B.3 ANALYSIS EXPERIMENT OF $w$ ON EDITING

We demonstrate the result of the analysis experiment of the parameter $w$ in our proposed `mimic-CFG` method. As shown in Fig. B.6, the result indicates that $w = 0.94$ provides the best balance between background preservation and editing quality for Euler and Heun, $w = 0.92$ provides the best balance between background preservation and editing quality for RF-Solver

## B.4 ANALYSIS EXPERIMENT OF $w$ ON STABLE DIFFUSION-3.5

We demonstrate the result of the analysis experiment of the parameter $w$ in our proposed `mimic-CFG` method. As shown in Fig. B.7, the result indicates that $w = 0.94$ provides the best balance between background preservation and editing quality

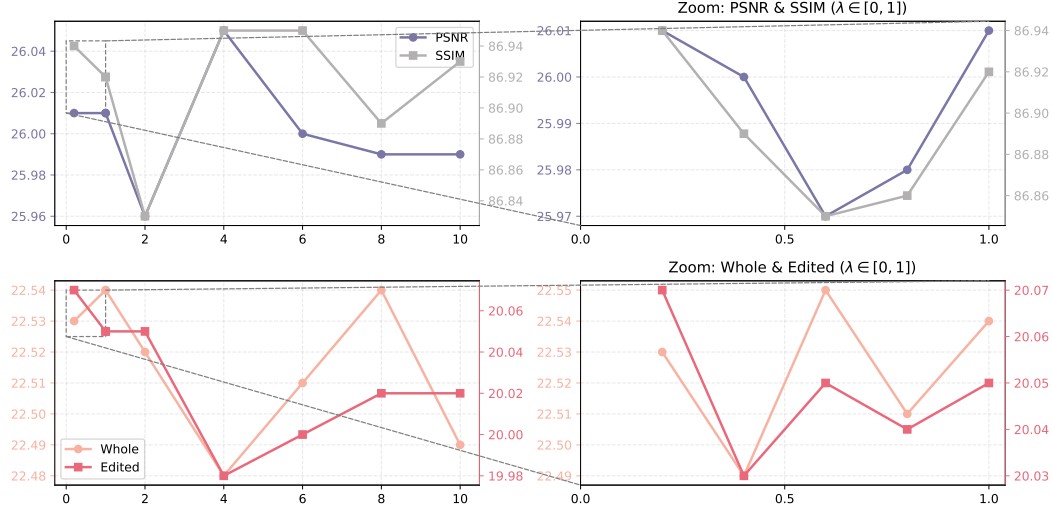

Figure B.4: Analysis experiment on the effect of the proximal operator parameter $\lambda$ with the solver **Heun**.

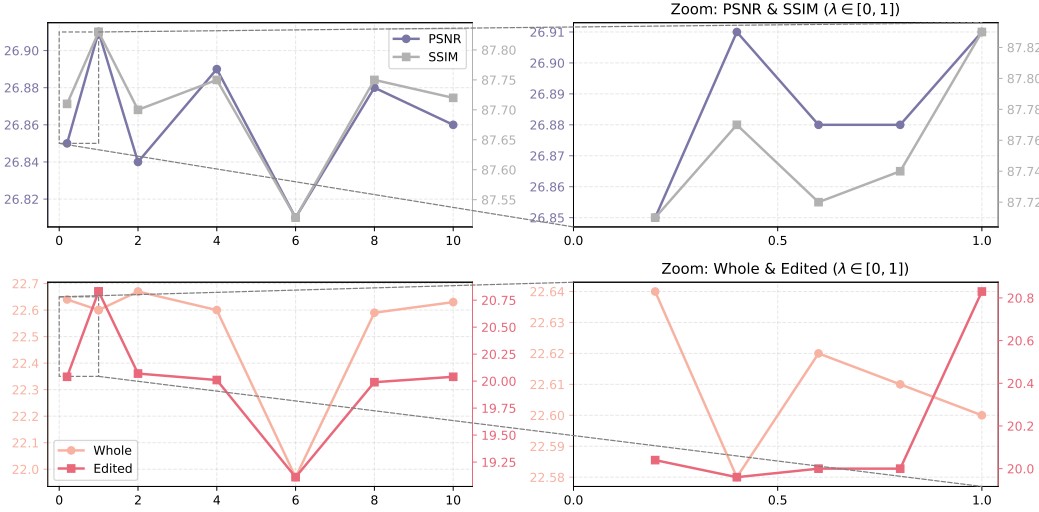

Figure B.5: Analysis experiment on the effect of the proximal operator parameter $\lambda$ with the solver **RF-Solver**.

## B.5 ANALYSIS EXPERIMENT OF $\lambda$ ON STABLE-DIFFUSION3.5

We demonstrate the result of the analysis experiment of the parameter $\lambda$ in our proposed `PMI` method. As shown in Fig. B.8 and Fig. B.9 the result indicates that $\lambda = 2$ provides the best performance on reconstruction task, and $\lambda = 1$ provides the best performance on editing task.

## B.6 ANALYSIS EXPERIMENT OF $\epsilon$

We demonstrate the result of the analysis experiment of the parameter $\epsilon$ in our proposed `PMI` method. As shown in Fig. B.10, the result indicates that $\epsilon = 2$ provides the best balance between background preservation and editing quality.

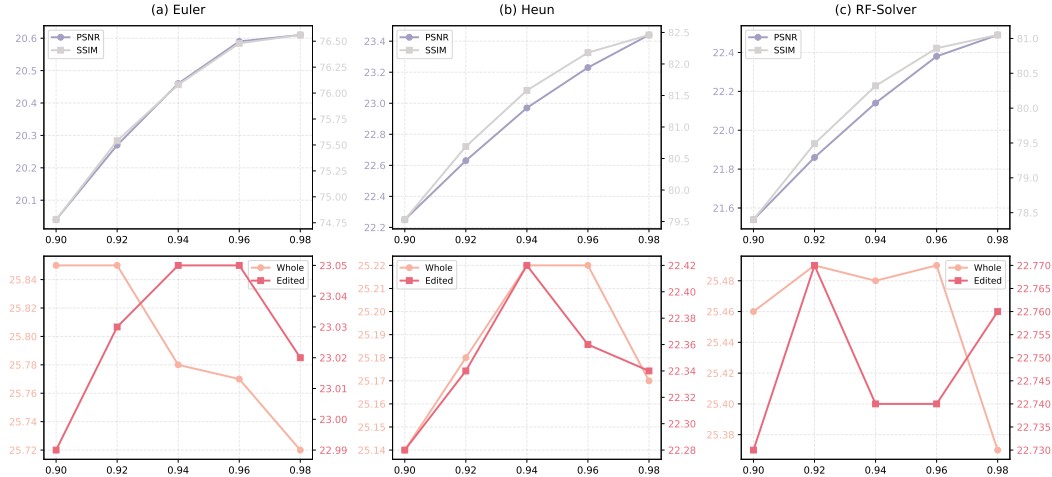

Figure B.6: Analysis experiment on the effect of the `mimic-CFG` parameter $w$ with three different solvers, including Euler, Heun, and RF-Solver.

Table B.1: Comparison on different velocity computing scheme. The results show that `PMI + mimic-CFG` achieve better results across all metrics in the editing task, especially in terms of CLIP similarity. Best results are in bold.

| Method | Structure Distance$^{\downarrow}_{\times 10^3}$ | Background Preservation | | | | CLIP Similarity | | NFE $\downarrow$ |
|---|---|---|---|---|---|---|---|---|
| | | PSNR $\uparrow$ | LPIPS$^{\downarrow}_{\times 10^3}$ | MSE$^{\downarrow}_{\times 10^4}$ | SSIM$^{\uparrow}_{\times 10^2}$ | Whole $\uparrow$ | Edited $\uparrow$ | |
| FireFlow | 29.39 | 22.40 | 136.49 | 79.35 | 80.78 | 25.35 | 22.52 | 18 |
| + EMA | 28.82 | 22.74 | 129.36 | 78.59 | **82.97** | 25.39 | 22.54 | 18 |
| + Ours | **28.01** | **22.79** | **128.55** | **78.02** | 81.76 | **25.50** | **22.68** | 18 |

## B.7 Different Velocity Averaging Scheme

To further stabilize the velocity accumulation used in our trajectory correction module, we introduce an Exponential Moving Average (EMA) formulation as an alternative to the simple summation-based mean velocity adopted in the Eq. equation 8.

The original scheme accumulates the velocity contribution across steps as

$$\bar{\mathrm{v}}_{t_k} = \bar{\mathrm{v}}_{t_{k-1}} + \Delta t_k \mathrm{v}_{t_k},$$

which may propagate noise due to the unweighted influence of all past increments. To obtain a smoother estimate of the running velocity, we adopt EMA, which gradually down-weights older increments via exponential smoothing.

Let $\mathrm{u}_{t_k} = \Delta t_k \mathrm{v}_{t_k}$ denote the velocity increment at step $k$. The EMA update is defined as

$$\mathrm{v}^{\mathrm{ema}}_{t_k} = \alpha\, u^k + (1 - \alpha)\, \mathrm{v}^{\mathrm{ema}}_{t_k},$$

where $\alpha \in (0, 1)$ controls the smoothing strength.

We compare EMA with the original integral-based averaging strategy in Table B.1. Without any parameter tuning, EMA already achieves improvements over the FireFlow baseline, indicating that smoothing-based averaging can be beneficial even with minimal design effort. Our final method, which uses the proposed integral-based correction, achieves the best results across most metrics. This suggests that our velocity formulation provides more effective control over trajectory stability. We leave a more systematic investigation of EMA and additional averaging strategies for future work.

## B.8 Comparison on Different Editing Method

Tab. B.2 presents the comparison results between `mimic-CFG` and `PMI` in the editing process. The results show that while `mimic-CFG` slightly outperforms `PMI` in terms of PSNR and SSIM, it

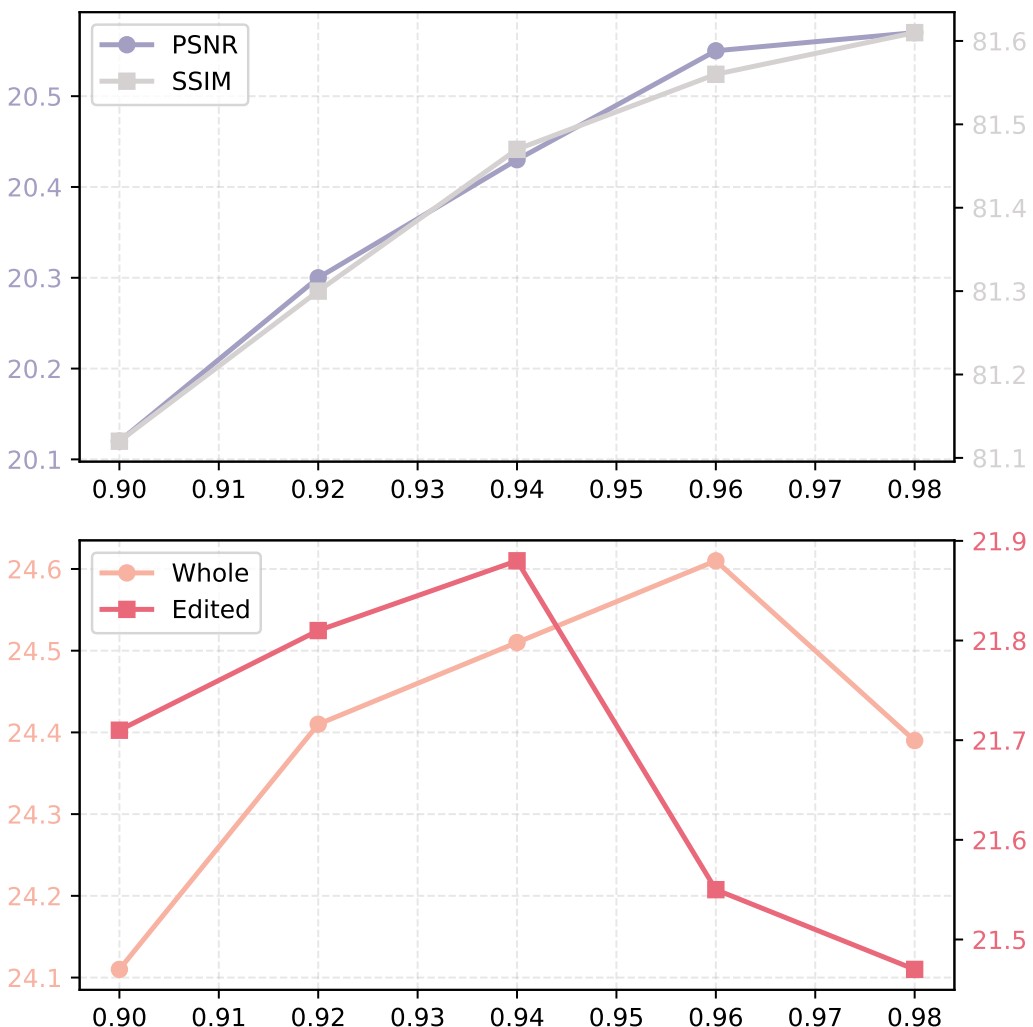

Figure B.7: Analysis experiment on the effect of the `mimic-CFG` parameter $w$ on **Stable Diffusion-3.5** with solver FireFlow.

Table B.2: Comparison on different editing method. The results show that `mimic-CFG` outperforms `PMI` across all metrics in the editing task, especially in terms of CLIP similarity. Best results are in bold.

| Method | Structure Distance$^{\downarrow}_{\times 10^3}$ | Background Preservation | | | | CLIP Similarity | | NFE $\downarrow$ | Time(s) |
|---|---|---|---|---|---|---|---|---|---|
| | | PSNR $\uparrow$ | LPIPS$^{\downarrow}_{\times 10^3}$ | MSE$^{\downarrow}_{\times 10^4}$ | SSIM$^{\uparrow}_{\times 10^2}$ | Whole $\uparrow$ | Edited $\uparrow$ | | |
| baseline | 29.39 | 22.40 | 136.49 | 79.35 | 80.78 | 25.35 | 22.52 | 18 | 5.1 |
| + PMI | 28.54 | 22.74 | 130.80 | 80.09 | 81.50 | 25.34 | 22.57 | 18 | 5.2 |
| + mimic-CFG | **28.01** | **22.79** | **128.55** | **78.02** | **81.76** | **25.50** | **22.68** | 18 | 5.2 |

significantly surpasses `PMI` in both CLIP Similarity scores, demonstrating its superior effectiveness in enhancing editing performance.

## B.9 QUANTITATIVE COMPARISON ON EDITING

Tab. B.3 demonstrates the comparison results on PIE-Bench with different NFE. We can observe that our method achieves results comparable to the baseline at higher NFE, even with significantly lower NFE. For example, Heun combined with our method achieves better structural consistency with fewer steps and lower NFE. Similarly, FireFlow benefits from our correction, showing higher

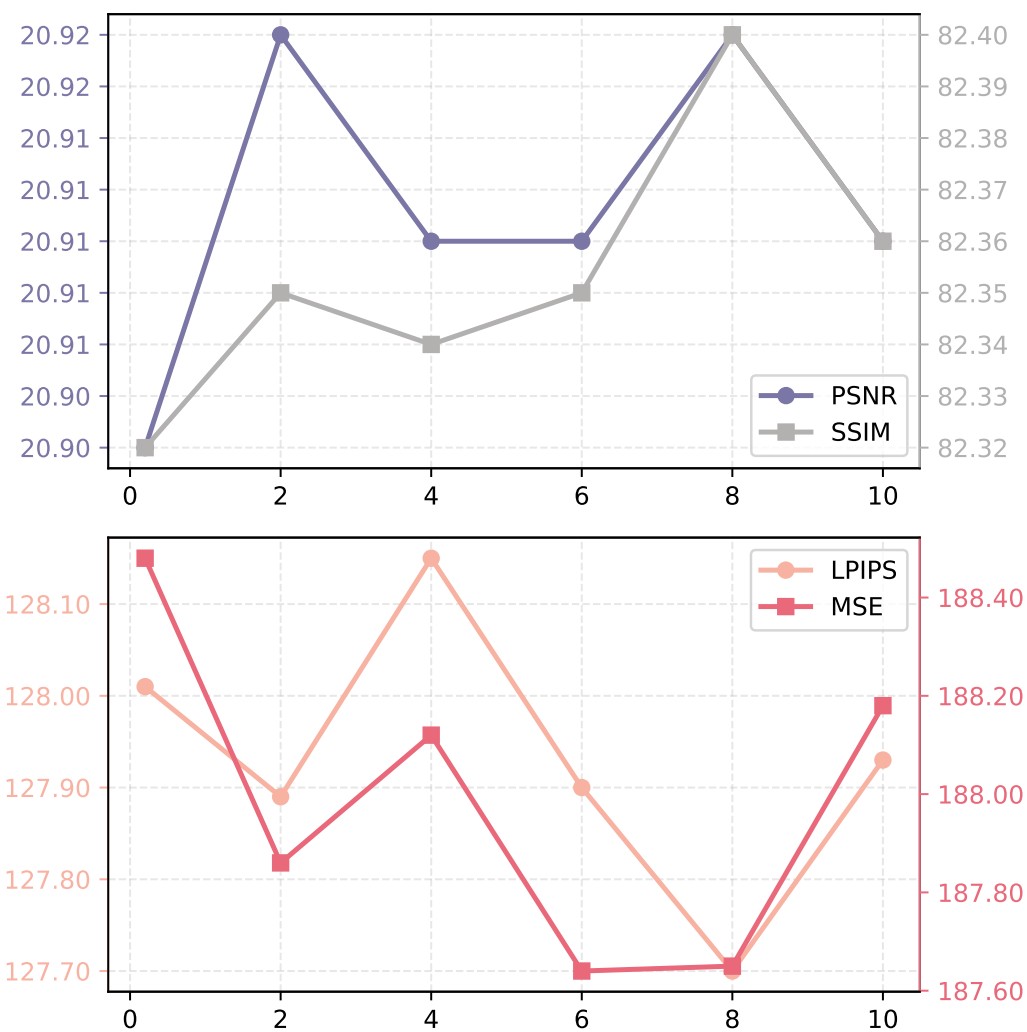

Figure B.8: Analysis experiment on the effect of the `PMI` parameter $\lambda$ on **Stable Diffusion-3.5** with solver FireFlow (Reconstruction).

Table B.3: Image editing comparison on PIE-Bench with different NFE. The results show that our method brings baseline metrics closer to those of high NFE baselines while reducing NFE, effectively accelerating the model. Best results are in bold and identical results are underlined.

| Method | Structure Distance$^{\downarrow}_{\times 10^3}$ | Background Preservation | | | | CLIP Similarity | | NFE $\downarrow$ |
|---|---|---|---|---|---|---|---|---|
| | | PSNR $\uparrow$ | LPIPS$^{\downarrow}_{\times 10^3}$ | MSE$^{\downarrow}_{\times 10^4}$ | SSIM$^{\uparrow}_{\times 10^2}$ | Whole $\uparrow$ | Edited $\uparrow$ | |
| Euler | **50.08** | **20.57** | **187.47** | 135.03 | 76.02 | 25.73 | **23.00** | 50 |
| + Ours | 52.09 | 20.36 | 193.21 | **121.59** | **76.11** | **25.89** | 22.79 | 40 |
| Heun | 33.27 | 22.52 | 142.17 | 93.71 | 80.99 | **25.34** | 22.27 | 60 |
| + Ours | **28.21** | **23.01** | **132.49** | **71.83** | **81.64** | 25.22 | **22.37** | 48 |
| RF-Inversion | **63.41** | 18.03 | 252.89 | 21.44 | 62.65 | 25.02 | 22.56 | 56 |
| + Ours | 64.34 | 18.03 | **252.85** | 21.44 | **62.94** | **25.38** | **22.76** | 52 |
| RF-Solver | 33.71 | 21.29 | 156.51 | 96.24 | 79.63 | 25.29 | 22.61 | 60 |
| + Ours | **32.13** | **22.35** | **146.40** | **81.45** | **80.46** | **25.44** | **22.64** | 52 |
| FireFlow | 34.30 | 21.91 | 147.68 | **96.71** | 80.16 | 25.56 | 22.67 | 26 |
| + Ours | **31.87** | **22.31** | **140.63** | 98.27 | **80.84** | **25.71** | **22.98** | 22 |

PSNR and better editing fidelity with reduced computational cost. Overall, our approach achieves

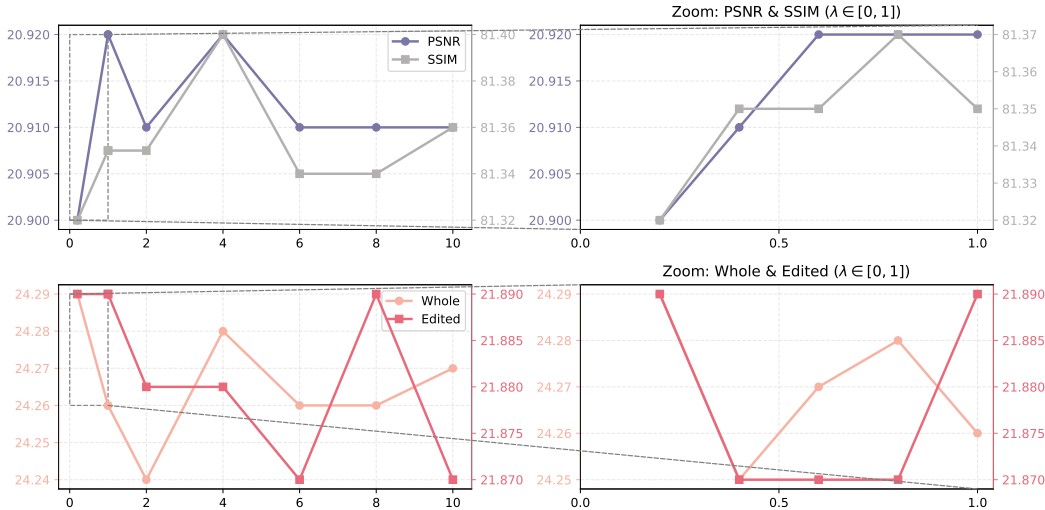

Figure B.9: Analysis experiment on the effect of the `PMI` parameter $\lambda$ on **Stable Diffusion-3.5** with solver FireFlow (Editing).

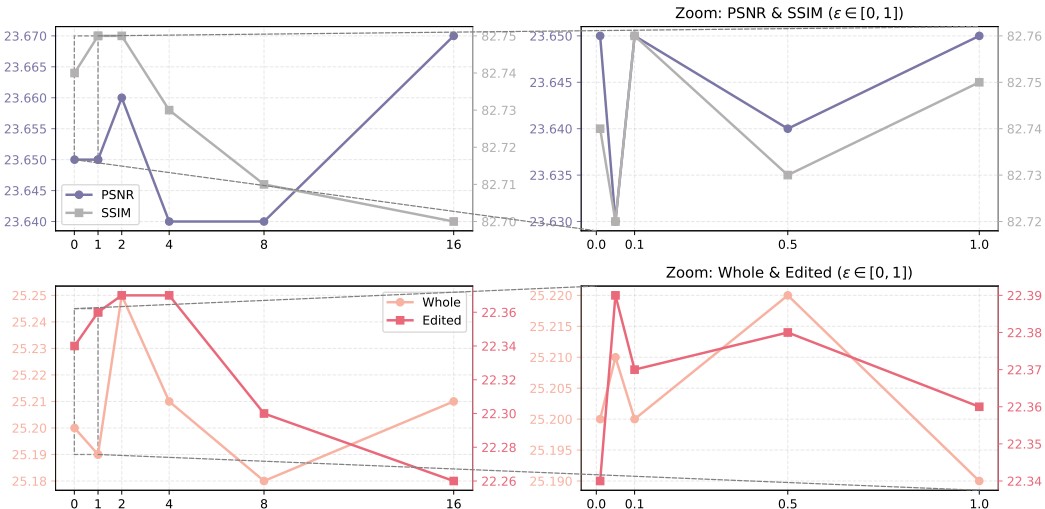

Figure B.10: Analysis experiment on the effect of the `PMI` parameter $\epsilon$ on FireFlow.

Table B.4: Editing performance on SDXL-Turbo. Our proximal correction significantly improves PSNR and SSIM while achieving the best CLIP similarity on both whole and edited regions.

| Method | Background Preservation | | | | CLIP Similarity | |
|---|---|---|---|---|---|---|
| | PSNR $\uparrow$ | LPIPS$_{\times 10^3}$ $\downarrow$ | MSE$_{\times 10^4}$ $\downarrow$ | SSIM$_{\times 10^2}$ $\uparrow$ | Whole $\uparrow$ | Edited $\uparrow$ |
| GNRI | 21.96 | 126.63 | 93.27 | 75.00 | 24.90 | 21.98 |
| Turbo-Edit | 22.26 | 107.30 | 99.96 | 80.05 | 25.85 | 22.74 |
| Direct Inv | 22.46 | 105.53 | 79.96 | 80.22 | 25.64 | 22.68 |
| DDIM | 22.00 | 118.56 | 90.16 | 76.00 | 24.38 | 21.48 |
| DDIM + Ours | **22.49**$_{0.49\uparrow}$ | **105.17** $_{13.9\downarrow}$ | **84.01** $_{6.15\downarrow}$ | **80.27** $_{4.27\uparrow}$ | **25.62**$_{1.24\uparrow}$ | **22.70** $_{1.22\uparrow}$ |

a more balanced trade-off between background preservation and editing quality while reducing the required sampling steps.

To verify the generalization capability of our method, we applied our proximal correction to a widely adopted diffusion model architecture. As shown in Tab. B.4, we incorporated our method into the

Table B.5: Reconstruction performance of SD3.5 with FireFlow.

| Method | PSNR↑ | LPIPS$_{\times 10^3}$ ↓ | MSE$_{\times 10^4}$ ↓ | SSIM$_{\times 10^2}$ ↑ |
|---|---|---|---|---|
| FireFlow | 20.91 | 128.16 | 189.79 | 82.32 |
| FireFlow + Ours | **20.92** | **127.89** | **187.86** | **82.35** |

Table B.6: Editing performance of SD3.5 with FireFlow.

| Method | PSNR↑ | LPIPS$_{\times 10^3}$ ↓ | MSE$_{\times 10^4}$ ↓ | SSIM$_{\times 10^2}$ ↑ | Whole↑ | Edited↑ |
|---|---|---|---|---|---|---|
| FireFlow | 20.57 | 150.73 | 137.14 | 81.52 | 24.51 | 21.71 |
| FireFlow + Ours | **20.67** | **148.63** | **135.43** | **81.76** | **24.58** | **21.83** |

DDIM sampler on the SDXL-Turbo. For comprehensive comparison, we also experimented with several state-of-the-art methods on SDXL-Turbo, including DDIM Song et al. (2022), GNRI Samuel et al. (2025) , Turbo-Edit Wu et al. (2024), and Direct Inv Ju et al. (2024). The results demonstrate that our method can be effectively applied to diffusion-based models.

We also applied our proximal correction to Stable Diffusion 3.5. As shown in Tab. B.6 and Tab. B.5, our method can be effectively applied to other RF-based model.

### B.10 RECONSTRUCTION ERROR

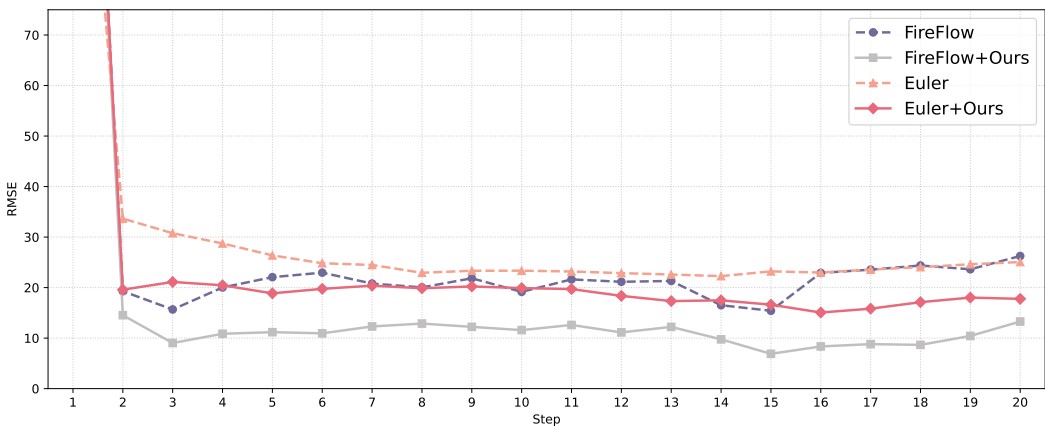

Figure B.11: Reconstruction error (RMSE) across different sampling steps. Our method consistently reduces error compared to FireFlow and Euler.

Fig. B.11 compares reconstruction errors under different solvers. FireFlow shows relatively stable errors, while FireFlow + Ours achieves consistently lower values across most steps. Euler suffers from error accumulation, especially at small step counts, whereas Euler + Ours effectively suppresses this growth and maintains lower reconstruction error throughout.

## C EXPERIMENTAL SETTINGS

### C.1 COMPUTING INFRASTRUCTURE.

The experiments in this paper were conducted using the following computing infrastructure:

- **Hardware:**
  - **GPU**: A single NVIDIA A100-PCIE-40GB with 40GB memory.
  - **CPU**: 10 vCPUs, Intel Xeon Processor (Skylake, IBRS).

- **Memory**: 72GB RAM.
- **Software:**
  - **Operating System**: Ubuntu 22.04.
  - **Python**: Python 3.12.
  - **PyTorch**: PyTorch 2.5.1.
  - **CUDA**: CUDA 12.4.

## C.2 INVERSION AND RECONSTRUCTION

We follow the official open-source implementation of `FireFlow`. For Euler, Heun, RF-Solver and FireFlow, the editing configuration see Tab. B.7.

Table B.7: Hyperparameter setting in Reconstruction task.

| Argument / Solver | Euler | Heun | RF-Solver | FireFlow |
|---|---|---|---|---|
| --guidance | 2 | 2 | 2 | 2 |
| --start_layer_index | 20 | 20 | 20 | 20 |
| --end_layer_index | 37 | 37 | 37 | 37 |
| --inject | 1 | 1 | 1 | 1 |
| --reuse_v | 1 | 1 | 1 | 1 |
| --editing_strategy | replace_v | replace_v | replace_v | replace_v |
| --qkv_ratio | 1.0,1.0,1.0 | 1.0,1.0,1.0 | 1.0,1.0,1.0 | 1.0,1.0,1.0 |

**Quantitative comparison.** For Tab. 1, all results are obtained using Flux.1. We use 30 Euler steps for the baseline, and 12 steps for Heun, RF-Solver, and FireFlow.

**Qualitative comparison.** For Fig. 1, we set the same number of function evaluations (NFE) for both conditional and unconditional reconstructions. All methods, with and without our velocity correction, are evaluated using $\mathbf{step} = 8$.

## C.3 EDITING

We again follow the official FireFlow implementation. For Euler, Heun, RF-Solver and FireFlow, the editing configuration see Tab. B.8.

Table B.8: Hyperparameter setting in Editing task.

| Argument / Method | Euler | Heun | RF-Solver | FireFlow |
|---|---|---|---|---|
| --guidance | 2 | 2 | 2 | 2 |
| --start_layer_index | 0 | 0 | 0 | 0 |
| --end_layer_index | 37 | 37 | 37 | 37 |
| --inject | 2 | 2 | 2 | 2 |
| --reuse_v | 1 | 1 | 1 | 1 |
| --editing_strategy | replace_v | replace_v | replace_v | replace_v |
| --qkv_ratio | 1.0,1.0,1.0 | 1.0,1.0,1.0 | 1.0,1.0,1.0 | 1.0,1.0,1.0 |

**Qualitative comparison.** For Fig. 2, RF-Inversion is evaluated with $\mathbf{step} = 26$ (Ours) and $\mathbf{step} = 28$ (Vanilla). RF-Solver uses $\mathbf{step} = 12$ (Ours) and $\mathbf{step} = 15$ (Vanilla). FireFlow uses $\mathbf{step} = 10$ (Ours) and $\mathbf{step} = 12$ (Vanilla).

## C.4 ABLATION STUDIES.

**Ablation study of $w$.** For ablation study of $w$, FireFlow is chosen as the baseline with step $= 12$ for all experiments in Section 5.4.

**Ablation study on the choice of norm.** For ablation study the choice of norm., FireFlow is chosen as the baseline with step $= 10$ for all experiments in Section 5.4.

---

**Algorithm 3** `PMI`: RF-Solver Case

**Input**: velocity $v_{\boldsymbol{\theta}}$, image $\mathbf{z}_0$, schedule $\mathcal{T} = \{t_0, \ldots, t_N\}$.
**Output**: $\hat{\mathbf{z}}_1$

1: $\mathbf{x}_{\text{dist}} = \mathbf{0}$
2: **for** $i = 0$ to $N - 1$ **do**
3: $\quad \Delta t = t_{i+1} - t_i$
4: $\quad \mathbf{v}_{t_i} = v_{\boldsymbol{\theta}}(\mathbf{z}_{t_i}, t_i)$
5: $\quad \mathbf{z}_{t_i}^{\text{mid}} = \mathbf{z}_{t_i} + \frac{1}{2}\Delta t\, \mathbf{v}_{t_i}$
6: $\quad \mathbf{v}_{t_i}^{\text{RFS}} = v_{\boldsymbol{\theta}}(\mathbf{z}_{t_i}^{\text{mid}}, \frac{1}{2}(t_{i+1} + t_i))$
7: $\quad \mathbf{x}_{\text{dist}} += \Delta t\, \mathbf{v}_{t_i}^{\text{RFS}}$
8: $\quad \bar{\mathbf{v}}_{t_i} = \frac{1}{t_{i+1} - t_0}\mathbf{x}_{\text{dist}}$
9: $\quad \hat{\mathbf{v}}_{t_i} = \mathbf{v}_{t_i}^{\text{RFS}} - r_{t_i}\frac{\nabla F(\mathbf{v}_{t_i}^{\text{RFS}})}{\|\nabla F(\mathbf{v}_{t_i}^{\text{RFS}})\|_2}$
10: $\quad \mathbf{a}_{t_i} = \dfrac{\hat{\mathbf{v}}_{t_i} - \mathbf{v}_{t_i}}{\Delta t/2}$
11: $\quad \mathbf{z}_{t_{i+1}} = \mathbf{z}_{t_i} + \Delta t\, \hat{\mathbf{v}}_{t_i} + \frac{1}{2}\Delta t^2\, \mathbf{a}_{t_i}$
12: **end for**
13: **return** $\mathbf{z}_1$

---

**Algorithm 4** `mimic-CFG`: RF-Solver Case

**Input**: velocity $v_{\boldsymbol{\theta}}$, image $\mathbf{z}_0$, schedule $\mathcal{T} = \{t_0, \ldots, t_N\}$, weight $w$.
**Output**: $\hat{\mathbf{z}}_0$

1: $\hat{\mathbf{z}}_1 = \text{PMI}(v_{\boldsymbol{\theta}}, \mathbf{z}_0, t)$
2: $\mathbf{x}_{\text{dist}} = \mathbf{0}$
3: **for** $i = N$ to $1$ **do**
4: $\quad \Delta t = t_{i-1} - t_i$
5: $\quad \mathbf{v}_{t_i} = v_{\boldsymbol{\theta}}(\hat{\mathbf{z}}_{t_i}, t_i)$
6: $\quad \mathbf{z}_{t_i}^{\text{mid}} = \hat{\mathbf{z}}_{t_i} + \frac{1}{2}\Delta t\, \mathbf{v}_{t_i}$
7: $\quad \mathbf{v}_{t_i}^{\text{RFS}} = v_{\boldsymbol{\theta}}(\mathbf{z}_{t_i}^{\text{mid}}, \frac{1}{2}(t_{i+1} + t_i))$
8: $\quad \mathbf{x}_{\text{dist}} += \Delta t\, \mathbf{v}_{t_i}^{\text{RFS}}$
9: $\quad \bar{\mathbf{v}}_{t_i} = \frac{1}{t_{i-1} - t_N}\mathbf{x}_{\text{dist}}$
10: $\quad \bar{\mathbf{v}}_{t_i}^{\text{proj}} = \dfrac{\mathbf{v}_{t_i}^{\text{RFS}\top}\bar{\mathbf{v}}_{t_i}^{\text{edit}}}{\|\bar{\mathbf{v}}_{t_i}^{\text{edit}}\|_2^2}\bar{\mathbf{v}}_{t_i}^{\text{edit}}$
11: $\quad \hat{\mathbf{v}}_{t_i} = (1 - w)\bar{\mathbf{v}}_{t_i}^{\text{proj}} + w\, \mathbf{v}_{t_i}^{\text{RFS}}$
12: $\quad \mathbf{a}_{t_i} = \dfrac{\hat{\mathbf{v}}_{t_i} - \mathbf{v}_{t_i}}{\Delta t/2}$
13: $\quad \mathbf{z}_{t_{i-1}} = \mathbf{z}_{t_i} + \Delta t\, \hat{\mathbf{v}}_{t_i} + \frac{1}{2}\Delta t^2\, \mathbf{a}_{t_i}$
14: **end for**
15: **return** $\hat{\mathbf{z}}_0$

---

**Algorithm 5** `PMI`: Heun Case

**Input**: velocity $v_{\boldsymbol{\theta}}$, image $\mathbf{z}_0$, schedule $\mathcal{T} = \{t_0, \ldots, t_N\}$.
**Output**: $\hat{\mathbf{z}}_1$

1: $\mathbf{x}_{\text{dist}} = \mathbf{0}$
2: **for** $i = 0$ to $N - 1$ **do**
3: $\quad \Delta t = t_{i+1} - t_i$
4: $\quad \mathbf{v}_{t_i} = v_{\boldsymbol{\theta}}(\mathbf{z}_{t_i}, t_i)$
5: $\quad \mathbf{z}_{t_i}^{\text{mid}} = \mathbf{z}_{t_i} + \frac{1}{2}\Delta t\, \mathbf{v}_{t_i}$
6: $\quad \mathbf{v}_{t_i}^{\text{heun}} = v_{\boldsymbol{\theta}}(\mathbf{z}_{t_i}^{\text{mid}}, \frac{1}{2}(t_{i+1} + t_i))$
7: $\quad \mathbf{x}_{\text{dist}} += \Delta t\, \mathbf{v}_{t_i}^{\text{heun}}$
8: $\quad \bar{\mathbf{v}}_{t_i} = \frac{1}{t_{i+1} - t_0}\mathbf{x}_{\text{dist}}$
9: $\quad \hat{\mathbf{v}}_{t_i} = \mathbf{v}_{t_i}^{\text{heun}} - r_{t_i}\frac{\nabla F(\mathbf{v}_{t_i}^{\text{heun}})}{\|\nabla F(\mathbf{v}_{t_i}^{\text{heun}})\|_2}$
10: $\quad \mathbf{z}_{t_{i+1}} = \mathbf{z}_{t_i} + \Delta t\, \hat{\mathbf{v}}_{t_i}$
11: **end for**
12: **return** $\mathbf{z}_1$

---

**Algorithm 6** `mimic-CFG`: Heun Case

**Input**: velocity $v_{\boldsymbol{\theta}}$, image $\mathbf{z}_0$, schedule $\mathcal{T} = \{t_0, \ldots, t_N\}$, weight $w$.
**Output**: $\hat{\mathbf{z}}_0$

1: $\hat{\mathbf{z}}_1 = \text{PMI}(v_{\boldsymbol{\theta}}, \mathbf{z}_0, t)$
2: $\mathbf{x}_{\text{dist}} = \mathbf{0}$
3: **for** $i = N$ to $1$ **do**
4: $\quad \Delta t = t_{i-1} - t_i$
5: $\quad \mathbf{v}_{t_i} = v_{\boldsymbol{\theta}}(\hat{\mathbf{z}}_{t_i}, t_i)$
6: $\quad \mathbf{z}_{t_i}^{\text{mid}} = \hat{\mathbf{z}}_{t_i} + \frac{1}{2}\Delta t\, \mathbf{v}_{t_i}$
7: $\quad \mathbf{v}_{t_i}^{\text{heun}} = v_{\boldsymbol{\theta}}(\mathbf{z}_{t_i}^{\text{mid}}, \frac{1}{2}(t_{i+1} + t_i))$
8: $\quad \mathbf{x}_{\text{dist}} += \Delta t\, \mathbf{v}_{t_i}^{\text{heun}}$
9: $\quad \bar{\mathbf{v}}_{t_i} = \frac{1}{t_{i-1} - t_N}\mathbf{x}_{\text{dist}}$
10: $\quad \bar{\mathbf{v}}_{t_i}^{\text{proj}} = \dfrac{\mathbf{v}_{t_i}^{\text{heun}\top}\bar{\mathbf{v}}_{t_i}^{\text{edit}}}{\|\bar{\mathbf{v}}_{t_i}^{\text{edit}}\|_2^2}\bar{\mathbf{v}}_{t_i}^{\text{edit}}$
11: $\quad \hat{\mathbf{v}}_{t_i} = (1 - w)\bar{\mathbf{v}}_{t_i}^{\text{proj}} + w\, \mathbf{v}_{t_i}^{\text{heun}}$
12: $\quad \hat{\mathbf{z}}_{t_{i-1}} = \hat{\mathbf{z}}_{t_i} + \Delta t\, \hat{\mathbf{v}}_{t_i}$
13: **end for**
14: **return** $\hat{\mathbf{z}}_0$

---

### C.5 HIGH-ORDER ALGORITHMS

For better reproducibility, we additionally provide the full implementations of `PMI` and `mimic-CFG` under several commonly used ODE solvers beyond the Euler case. These variants share the same correction structure while differing in the underlying base velocity estimation. Specifically, Algorithm 5 and Algorithm 6 present the Heun implementation, Algorithm 3 and Algorithm 4 correspond to the RF-Solver version, and Algorithm 7 together with Algorithm 8 show

the FireFlow integration. We hope these complete solver-specific implementations facilitate further research and replication of our results.

---

**Algorithm 7** `PMI`: FireFlow Case

---

**Input**: velocity function $v_{\boldsymbol{\theta}}(\cdot)$, image $\mathbf{z}_0$, time schedule $\mathcal{T} = \{t_0 = 0, \ldots, t_N = 1\}$.
**Output**: $\hat{\mathbf{z}}_1$

1: $\mathbf{x}_{\text{dist}} = \mathbf{0}$
2: **for** $i = 0$ to $N - 1$ **do**
3:     **if** i=0 **then**
4:         $\hat{\mathbf{v}}_{t_i}^{\text{mid}} = v_{\boldsymbol{\theta}}(\mathbf{z}_{t_i}, t_i)$
5:     **else**
6:         $\hat{\mathbf{v}}_{t_i}^{\text{mid}} = \mathbf{v}^{\text{previous}}$
7:     **end if**
8:     $\mathbf{z}_{t_i}^{\text{mid}} = \mathbf{z}_{t_i} + \frac{1}{2}\Delta t\, \hat{\mathbf{v}}_{t_i}^{\text{mid}}$
9:     $\mathbf{v}_{t_i} = v_{\boldsymbol{\theta}}(\mathbf{z}_{t_i}^{\text{mid}}, \frac{1}{2}(t_{i+1} + t_i))$
10:    $\mathbf{x}_{\text{dist}} += (t_{i+1} - t_i)\mathbf{v}_{t_i}$
11:    $\bar{\mathbf{v}}_{t_i} = \dfrac{1}{t_{i+1} - t_0}\mathbf{x}_{\text{dist}}$
12:    $\hat{\mathbf{v}}_{t_i} = \mathbf{v}_{t_i} - r_{t_i}\dfrac{\nabla F(\mathbf{v}_{t_i})}{\|\nabla F(\mathbf{v}_{t_i})\|_2}$
13:    $\mathbf{v}^{\text{previous}} = \hat{\mathbf{v}}_{t_i}$
14:    $\hat{\mathbf{z}}_{t_{i+1}} = \hat{\mathbf{z}}_{t_i} + (t_{i+1} - t_i)\hat{\mathbf{v}}_{t_i}$
15: **end for**
16: **return** $\hat{\mathbf{z}}_1$

---

**Algorithm 8** `mimic-CFG`: FireFlow Case

---

**Input**: velocity function $v_{\boldsymbol{\theta}}(\cdot)$, image $\mathbf{z}_0$, time schedule $\mathcal{T} = \{t_0 = 0, \ldots, t_N = 1\}$, interpolation weight $w$.
**Output**: $\hat{\mathbf{z}}_0$

1: $\hat{\mathbf{z}}_1 = \text{PMI}(v_{\boldsymbol{\theta}}, \mathbf{z}_0, t)$
2: $\mathbf{x}_{\text{dist}} = \mathbf{0}$
3: **for** $i = N$ to 1 **do**
4:     **if** i=N **then**
5:         $\hat{\mathbf{v}}_{t_i}^{\text{mid}} = v_{\boldsymbol{\theta}}(\mathbf{z}_{t_i}, t_i)$
6:     **else**
7:         $\hat{\mathbf{v}}_{t_i}^{\text{mid}} = \mathbf{v}^{\text{previous}}$
8:     **end if**
9:     $\mathbf{z}_{t_i}^{\text{mid}} = \mathbf{z}_{t_i} + \frac{1}{2}\Delta t\, \hat{\mathbf{v}}_{t_i}^{\text{mid}}$
10:    $\mathbf{v}_{t_i} = v_{\boldsymbol{\theta}}(\mathbf{z}_{t_i}^{\text{mid}}, \frac{1}{2}(t_{i-1} + t_i))$
11:    $\mathbf{x}_{\text{dist}} += (t_{i-1} - t_i)\mathbf{v}_{t_i}$
12:    $\bar{\mathbf{v}}_{t_i}^{\text{edit}} = \dfrac{1}{t_{i-1} - t_N}\mathbf{x}_{\text{dist}}$
13:    $\bar{\mathbf{v}}_{t_i}^{\text{proj}} = \dfrac{\mathbf{v}_{t_i}^{\top}\bar{\mathbf{v}}_{t_i}^{\text{edit}}}{\|\bar{\mathbf{v}}_{t_i}^{\text{edit}}\|_2^2}\bar{\mathbf{v}}_{t_i}^{\text{edit}}$
14:    $\hat{\mathbf{v}}_{t_i} = (1 - w)\bar{\mathbf{v}}_{t_i}^{\text{proj}} + w\,\mathbf{v}_{t_i}$
15:    $\mathbf{v}^{\text{previous}} = \hat{\mathbf{v}}_{t_i}$
16:    $\hat{\mathbf{z}}_{t_{i-1}} = \hat{\mathbf{z}}_{t_i} + (t_{i-1} - t_i)\hat{\mathbf{v}}_{t_i}$
17: **end for**
18: **return** $\hat{\mathbf{z}}_0$

---

# D   ADDITIONAL QUALITATIVE COMPARISON

## D.1   QUALITATIVE COMPARISON FOR EDITING

We also demonstrate more editing modalities in PIE-Bench, as shown in Fig. D.1, our method helps baselines achieve better result on Change Content, Change Pose, and Change Material. The row in Fig. D.1 is generated from RF-Solver, second row from FireFlow, and the third row from Euler.

# E   THE USE OF LARGE LANGUAGE MODELS (LLMS)

In accordance with the ICLR guidelines, we report the usage of large language models (LLMs) in the preparation of this paper. LLMs were employed solely as a writing assistance tool for language polishing, grammar correction, and improving readability. They were not used for research ideation, experimental design, data analysis, or the generation of scientific content. All technical ideas, theoretical developments, proofs, and experimental results presented in this paper are the work of the authors. The authors take full responsibility for the final content of the submission.

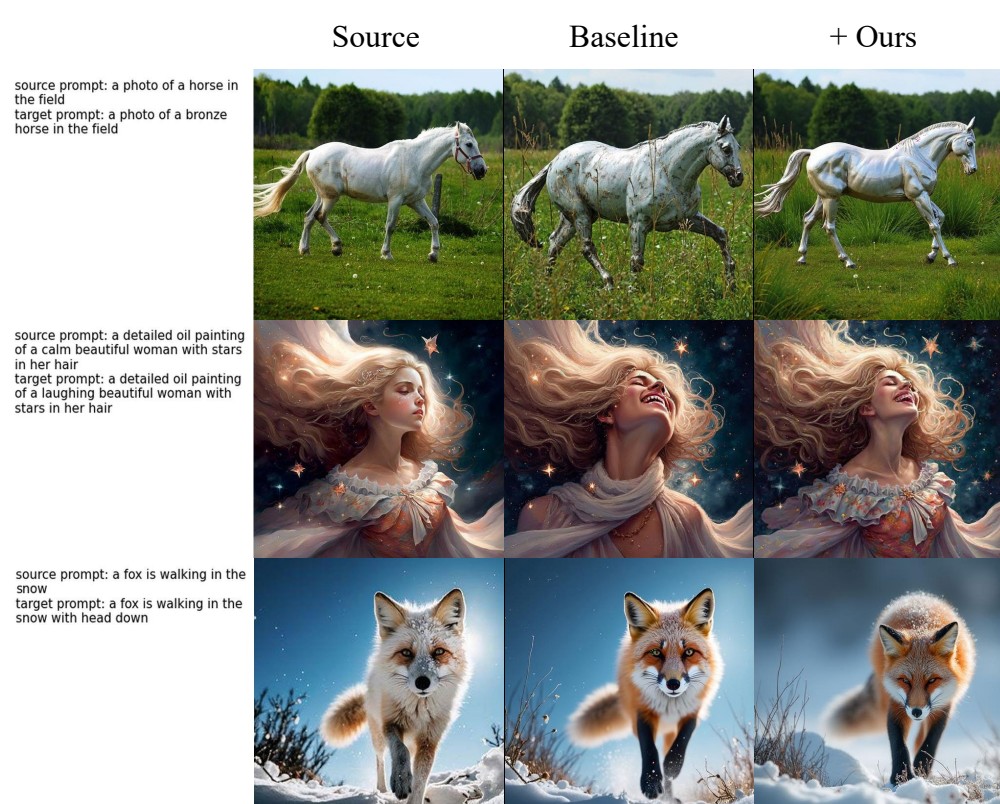

Figure D.1: Qualitative comparison of editing results on PIE-Bench. We considered more editing modalities in PIE-Bench, including Change Content, Change Pose, and Change Material. Our method help baseline achieve better results.

