# OpenReview forum: "Free Lunch for Stabilizing Rectified Flow Inversion"
_ICLR.cc/2026/Conference — ICLR 2026 Poster_

### Official Review · Reviewer_g18M · 2025-10-27

**Soundness:** 2
**Presentation:** 2
**Contribution:** 3
**Rating:** 6
**Confidence:** 5

**Summary:**

The paper proposes Proximal-Mean Inversion (PMI), a training-free technique designed to stabilize the velocity field during the inversion process of Rectified Flow (RF) models. The core idea is to apply a proximal operator that pulls the predicted current velocity towards a running average of past velocities ($\bar{v}_{t_{k}}$). This correction is constrained within a theoretically derived spherical Gaussian to prevent trajectory deviation into low-density regions. Furthermore, the paper introduces mimic-CFG for the editing stage, which uses an interpolation strategy (akin to Classifier-Free Guidance) between the current velocity and its projection onto the mean velocity direction to balance editability and structure preservation. The authors claim state-of-the-art performance in reconstruction and editing tasks on PIE-Bench with enhanced efficiency (fewer NFEs).

**Strengths:**

Strong Theoretical Motivation and Novelty: The identification of accumulated approximation errors leading to velocity field instability is a crucial problem in RF inversion. The proposed solution of using a running mean velocity for proximal correction is novel and intuitively sound, as it leverages the global consistency property of the Rectified Flow trajectory (near-constant velocity field).Zero NFE Overhead: The implementation of PMI as a correction step without requiring any additional Neural Function Evaluations (NFEs) is a significant practical advantage. Many existing inversion refinement methods (like FPI) achieve accuracy at the cost of increased computational load, while PMI promises improved stability for free in terms of NFE count.Theoretical Constraint and Justification: The derivation of the Stability Condition (Proposition 1), which defines the radius $r_i$ for the spherical Gaussian constraint, is a strong point. It provides a principled way to control the correction magnitude, tethering the solution to high-density regions based on instability theory (citing Zhang et al., 2025). This moves the method beyond a simple heuristic.Comprehensive Experimental Validation: The method is evaluated rigorously across four distinct solvers (Euler, Heun, RF-Solver, FireFlow) for both reconstruction and editing tasks, covering a wide range of integration complexity. The consistent quantitative improvements in PSNR, SSIM, and LPIPS across all baselines (Table 1 and Table 2) convincingly demonstrate the general effectiveness and plug-and-play nature of PMI.Effective Editing Strategy (mimic-CFG): The mimic-CFG strategy is a clever application of the PMI principle to the editing context, successfully adapting the highly effective CFG framework using only the internal, historically averaged velocity as the "unconditional" signal. This efficiently addresses the trade-off between editability and consistency.

**Weaknesses:**

1. Reliance on Simplistic Approximation in PMI:The core update (Eq. 13) is derived by minimizing a first-order Taylor approximation of $F(v)$ (Eq. 11) constrained by an $L_2$ sphere (Eq. 12). This reduces the complex proximal minimization to a simple gradient step, making the "Proximal" in "Proximal-Mean" a somewhat generous term. The simplicity is beneficial for efficiency, but the theoretical depth of the final step is thin.

2. The $\mathcal{O}(\Delta t_i^2)$ Error Analysis (Proposition 2) is Misleading:The paper claims the local error remains $\mathcal{O}(\Delta t_i^2)$, the same as the standard Euler method. This is expected, as PMI is a correction applied after the Euler step, and it does not change the core differential equation integration method. The proof is trivial and does not justify the benefit of PMI. The true benefit lies in correcting the accumulated errors by leveraging the running mean, not in improving the local truncation error bound, which is solely dictated by the Euler method's order. This proposition adds little value and distracts from the method's actual contribution.

3. Lack of Sensitivity Analysis for $r_{t_{k}}$ (Proposition 1):The Stability Condition is derived based on the full-trajectory Gaussian properties of $\hat{z}_1$. However, the final parameter choice for $r_i$ (Eq. 14) still includes an arbitrary $\epsilon$ term (line 793, 794) and is proportional to $\Delta t_i$. The paper provides no empirical analysis on how $\epsilon$ is chosen or how sensitive the results are to the constant factor $\sqrt{2n+3\sqrt{2n}}/T$. This makes the supposedly "theoretically derived" radius look like a complex, slightly obfuscated heuristic, undermining the strong theoretical claim.Limited


4. Editing Evaluation:While the quantitative results are strong, the editing evaluation is primarily focused on background preservation (PSNR, SSIM on unedited regions) and CLIP similarity (a weak proxy for true edit fidelity). A critical review requires evidence that the editing is semantically better and not just more consistent with the background. More focused metrics, such as a user study or more fine-grained editing task failures, would be required to fully validate the claimed "enhanced editing quality." The qualitative results in Figure 2 show subtle improvements, but the editing changes are often minimal.

**Questions:**

above

---

> ### Author Response · Authors · 2025-11-22
>
> **Q1**: Reliance on Simplistic Approximation in PMI:The core update (Eq. 13) is derived by minimizing a first-order Taylor approximation of $F(v)$ (Eq. 11) constrained by an $L_2$ sphere (Eq. 12). This reduces the complex proximal minimization to a simple gradient step, making the "Proximal" in "Proximal-Mean" a somewhat generous term. The simplicity is beneficial for efficiency, but the theoretical depth of the final step is thin.
>
> **A1**:
>
> Thank you for pointing this out. We agree that the current proximal derivation is overly simplified and does not fully reflect the depth suggested by the formulation. At the same time, this simplification also brings practical advantages: the first-order approximation yields a computationally lightweight and stable update direction. Moreover, as we further analyze in the revised appendix (**Appendix A.4**), incorporating second- or higher-order Taylor expansions does not change the optimal update direction; the quadratic term contributes only a constant Hessian ($\nabla^2 F(\mathbf{v}\_{t_k})=\frac{1}{\lambda}\mathbf{I}$), and all higher-order derivatives vanish ($\nabla^n F(\mathbf{v}\_{t_k}) = \mathbf{0}$), resulting in the same solution as the first-order case. This confirms that the simplified form is not an arbitrary reduction but a consequence of the objective’s structure. The resulting update provides a controlled correction that effectively reduces long-term drift under strict NFE constraints. These properties enable the method to deliver substantial empirical improvements over strong baselines despite its simple analytic form.
>
> **Q2**: The  $\mathcal{O}(\Delta t^2)$ Error Analysis (Proposition 2) is Misleading:The paper claims the local error remains $\mathcal{O}(\Delta t^2)$
> , the same as the standard Euler method. This is expected, as PMI is a correction applied after the Euler step, and it does not change the core differential equation integration method. The proof is trivial and does not justify the benefit of PMI. The true benefit lies in correcting the accumulated errors by leveraging the running mean, not in improving the local truncation error bound, which is solely dictated by the Euler method's order. This proposition adds little value and distracts from the method's actual contribution.
>
> **A2**:
>
> We thank the reviewer for the helpful observation. We agree that the local truncation error of the Euler solver remains $\mathcal{O}(\Delta t^2)$, and that a post–step correction cannot change the numerical order of the underlying ODE integrator. We also acknowledge that, in its current form, Proposition 2 may give the unintended impression that it aims to demonstrate an improved numerical order, which is not the intent.
> To avoid distractions from the main contribution, we move Proposition 2 to the appendix and revise its presentation.  As the reviewer correctly noted, the true benefit of PMI lies in reducing the accumulated integration errors along the entire inversion trajectory by leveraging the running mean, rather than improving the single–step truncation error.
> We appreciate the reviewer’s feedback and revise the manuscript so that this clarification is explicit and better aligned with the conceptual motivation of PMI.

---

> ### Author Response · Authors · 2025-11-22
>
> **Q3**: Lack of Sensitivity Analysis for $r_{t_k}$ (Proposition 1): The Stability Condition is derived based on the full-trajectory Gaussian properties of $\hat{\mathbf{z}}_1$. However, the final parameter choice for $r_i$ (Eq. 14) still includes an arbitrary $\epsilon$ term (line 793, 794) and is proportional to $\Delta t_i$. The paper provides no empirical analysis on how $\epsilon$ is chosen or how sensitive the results are to the constant factor $\sqrt{2n} + 3\sqrt{2n}/T$. This makes the supposedly "theoretically derived" radius look like a complex, slightly obfuscated heuristic, undermining the strong theoretical claim. Limited.
>
> **A3**:
>
> We thank the reviewer for the thoughtful comment. Proposition 1 is intended to provide a theoretical upper bound on the neighborhood radius and to justify that the PMI update remains within a stable region along the trajectory. We agree that the bound contains a constant factor parameterized by $\varepsilon$, which reflects the looseness inherent in the Gaussian concentration inequality used in the derivation. The purpose of this term is to ensure a valid probabilistic guarantee rather than to prescribe an exact numerical value for the radius.
> To address the reviewer’s concern regarding sensitivity, we conducted additional experiments analyzing $\varepsilon$. The results, included in **Appendix B.6 (Fig.B.7.)** show that the effective performance range is broad. While the theoretical constant factor is used as the default setting, the performance of PMI remains robust even when the constant is varied by a factor of up to 10x, confirming that the method does not rely on a hyper-optimized value of the theoretical bound.
>
> **Q4**:
> Editing Evaluation:While the quantitative results are strong, the editing evaluation is primarily focused on background preservation (PSNR, SSIM on unedited regions) and CLIP similarity (a weak proxy for true edit fidelity). A critical review requires evidence that the editing is semantically better and not just more consistent with the background. More focused metrics, such as a user study or more fine-grained editing task failures, would be required to fully validate the claimed "enhanced editing quality." The qualitative results in Figure 2 show subtle improvements, but the editing changes are often minimal.
>
> **A4**:
>
> We thank the reviewer for the valuable comment. The metrics used in our editing evaluation mainly follow prior works [1, 2]. To further demonstrate the effectiveness of our method beyond background consistency, we additionally include **Fig. D2**, showcasing more challenging editing modalities, including Change Content, Change Pose, and Change Material. In these settings, baseline methods show limited editing results, while our approach successfully completes the intended edits, indicating a clear improvement in semantic editing capability.
>
> [1] "Taming rectified flow for inversion and editing", ICML 2025
>
> [2] "Fireflow: Fast inversion of rectified flow for image semantic editing", ICML 2025

---

> ### Author Response · Authors · 2025-11-26
>
> Dear Reviewer,
>
> I hope this message finds you well. We hope we've addressed your concerns. If you have any further questions, please let us know. Your insights are invaluable to us, and we're eager to address any remaining issues to improve our work.
>
> Thank you for your time and effort in reviewing our paper.

---

> ### Comment · Reviewer_g18M · 2025-11-27
> **Feedback to Authors**
>
> Thank you for providing the supplementary explanations.
>
> While some of my inquiries have been addressed, I am still concerned about the proposed method. I will spend more time analyzing the contributions of this paper to the academic community in greater detail. I am also happy to discuss these issues with other reviewers and members of the academic committee.
>
> I decided to improve the rate between 4 to 6, which is 5.5, borderline accept.

---

> > ### Author Response · Authors · 2025-11-28
> > **Follow-up on Your Updated Comment in the Revision History**
> >
> > Dear Reviewer g18M,
> >
> > We noticed in the revision history that your comment has been updated, and we would like to sincerely thank you for taking the time to revisit our submission.
> >
> > We are glad to see that the earlier drop from **the original rating of 6 to 4** was due to a misunderstanding, and we appreciate your latest clarification indicating that you intend to **adjust the score back toward your intended evaluation**.
> >
> > If there are any remaining concerns or specific points you would like us to further address during the discussion phase, please let us know, and we would be very happy to continue the discussion and provide any additional clarification.
> >
> > We truly appreciate your thoughtful feedback and suggestions, which help us improve the quality and clarity of our work.
> >
> > Best regards,
> >
> > The authors

---

### Official Review · Reviewer_eSEF · 2025-10-30

**Soundness:** 2
**Presentation:** 3
**Contribution:** 2
**Rating:** 4
**Confidence:** 5

**Summary:**

This paper proposes Proximal-Mean Inversion (PMI), a training-free, gradient-based correction method for stabilizing velocity fields during inversion in rectified-flow (RF)-based generative models. The paper further introduces mimic-CFG, a velocity correction mechanism for editing tasks inspired by classifier-free guidance. Both methods aim to address approximation errors and instability in existing inversion approaches, improving reconstruction quality and editing fidelity. Experiments illustrate the effectiveness of the proposed methods.

**Strengths:**

- The motivation of this paper is clear. The inaccuracy of inversion is a noticeable problem of RF-based methods, where many prior works has been attempting to eliminate this. The proposed method aims to address this problem through gradient correction, which is intuitive and promising.
- The derivation of PMI is mathematically detailed, including closed-form updates (see Proposition 1 and Appendix A.1), and analysis of error bounds are also provided (Proposition 2, Appendix A.3).
- Experiments are comprehensive, demonstrating the effectiveness of the proposed method on various settings.
- The paper is well-written and easy to follow.

**Weaknesses:**

- Some ablation studies are missed. As stated in Line 238, authors use the first-order Taylor Expansion to estimate the objective in Equation (10). Authors are expected to conduct experiments to demonstrate how does the expansion order influence the performance.

- The settings of baseline methods is not clearly stated in the paper. Some hyperparameters in baseline methods are crucial to their performance (such as the feature sharing choices in FireFlow and RF-Solver). Empirically, the flaws of baseline methods in Figure 2 might be addressed through adjusting these hyperparameters. As a result, authors are expected to provide the more detailed information for fair comparion.

- Personally, I think the performance improvement of PMI is marginal, especially in the image reconstruction tasks. As illustrated in Figure 1, in the conditional case, the performance of RF-Solver is good enough. Although the unconditional image reconstruction results of PMI in Figure 1 are better, I think it is not necessary to test the performance under this setting because nowadays the description of images are easy to be obtained (for example, by MLLM), which can be served as the condition.

**Questions:**

See weakness

---

> ### Author Response · Authors · 2025-11-22
>
> **Q1**: Some ablation studies are missed. As stated in Line 238, authors use the first-order Taylor Expansion to estimate the objective in Equation (10). Authors are expected to conduct experiments to demonstrate how does the expansion order influence the performance.
>
> **A1**: Thank you for the insightful comment. We agree that studying the effect of Taylor expansion order is important for understanding the behavior of PMI. In the revised manuscript, we have added a detailed analysis of higher-order Taylor expansions in **Appendix A.4**, where we show that second-order and higher-order expansions lead to the same optimization direction as the first-order case due to the structure of the proximal objective. The analysis is as follows;
>
> ## **Analysis on the Expansion Order of $F(\mathbf{v})$**
>
> To better understand the effect of higher-order approximations in PMI, we analyze the Taylor expansion of the proximal objective $F(\mathbf{v})$ around the current velocity $\mathbf{v}_{t_k}$.
>
> ---
>
> ## **Second-order expansion**
>
> The second-order Taylor expansion of $F(\mathbf{v})$ around $\mathbf{v}_{t_k}$ is:
>
> $F(\mathbf{v})$ = $F(\mathbf{v}\_{t_k})$ + $\nabla F(\mathbf{v}\_{t_k})^\top (\mathbf{v}-\mathbf{v}\_{t_k})$ + $\frac{1}{2}(\mathbf{v}-\mathbf{v}\_{t_k})^\top\nabla^2 F(\mathbf{v}\_{t_k})(\mathbf{v}-\mathbf{v}\_{t_k})$
>
>
> Following the constrained formulation in Eq. (12):
>
> $\arg\min\_{|\mathbf{d}|\_2^2 = 1}$[$\nabla F(\mathbf{v}\_{t_k})^\top (\mathbf{v}-\mathbf{v}\_{t_k})$ + $\frac{1}{2}(\mathbf{v}-\mathbf{v}\_{t_k})^\top\nabla^2 F(\mathbf{v}\_{t_k})(\mathbf{v}-\mathbf{v}\_{t_k})]$
>
> Since $F$ contains an $\mathcal{L}\_1$ regularization term and a quadratic term, its Hessian is:
>
> $$
> \nabla^2 F(\mathbf{v}_{t_k})=
> \frac{1}{\lambda}\mathbf{I},
> $$
> Plugging this into the objective:
>
> $\arg\min\_{|\mathbf{d}|\_2^2 = 1}$$[\nabla F(\mathbf{v}\_{t_k})^\top (\mathbf{v}-\mathbf{v}\_{t_k})+\frac{1}{2\lambda}|\mathbf{v}-\mathbf{v}\_{t_k}|_2^2]$
>
> Using the spherical constraint:
>
> $$
> \mathbf{v} = \mathbf{v}_{t_k} + r\mathbf{d},
> \qquad |\mathbf{d}|_2 = 1,
> $$
>
> we obtain:
>
> $$
> \arg\min\_{|\mathbf{d}|\_2^2 = 1}
> \left[
> r,\nabla F(\mathbf{v}\_{t_k})^\top \mathbf{d}
> +
> \frac{r^2}{2\lambda}
> \right].
> $$
>
> Since the second term is independent of $\mathbf{d}$, the optimization reduces to:
>
> $$
> \arg\min \_ {|\mathbf{d}|\_2^2 = 1}
> r \nabla F(\mathbf{v}\_{t _ k})^\top \mathbf{d},
> $$
>
> which is **identical to the first-order case**.
>
> Thus, the optimal update direction remains unchanged even when second-order information is included.
>
> ---
>
> ## **Higher-order expansions**
>
> For $n \ge 3$, the Taylor expansion introduces higher-order derivatives:
>
> $$
> \nabla^n F(\mathbf{v}_{t_k}) = \mathbf{0},
> \qquad \forall n \ge 3.
> $$
>
> Therefore, all higher-order Taylor terms vanish.
>
> Thus, **third-order and higher-order expansions produce the same optimization objective as the second-order case**, yielding **the same update direction**.

---

> ### Author Response · Authors · 2025-11-22
>
> **Q2**: The settings of baseline methods is not clearly stated in the paper. Some hyperparameters in baseline methods are crucial to their performance (such as the feature sharing choices in FireFlow and RF-Solver). Empirically, the flaws of baseline methods in Figure 2 might be addressed through adjusting these hyperparameters. As a result, authors are expected to provide the more detailed information for fair comparion.
>
> **A2**:
>
> We thank the reviewer for pointing out the importance of clearly describing the hyperparameter settings of baseline methods. We agree that a fair comparison requires full transparency.
> In the revised manuscript, we have added a comprehensive description of all baseline settings in **Appendix C**, including the specific choices for feature sharing in FireFlow and RF-Solver,, and other critical parameters. We used the official, publicly released configurations from Fireflow for all baselines to ensure that our comparisons are based on the methods' best-performing, standard settings. The parcial settings are in tables below.
>
> ## Reconstruction
>
> | Argument / Solver       | Euler | Heun | RF-Solver | FireFlow |
> |-------------------------|:-----:|:----:|:---------:|:--------:|
> | `--guidance`            |   2   |   2  |     2     |    2     |
> | `--start_layer_index`   |  20   |  20  |    20     |   20     |
> | `--end_layer_index`     |  37   |  37  |    37     |   37     |
> | `--inject`              |   2   |   2  |     2     |    2     |
> | `--reuse_v`             |   1   |   1  |     1     |    1     |
> | `--editing_strategy`    | `replace_v` | `replace_v` | `replace_v` | `replace_v` |
> | `--qkv_ratio`           | `1.0,1.0,1.0` | same | same | same |
>
> ## Editing
>
> | Argument / Method      | Euler | Heun | RF-Solver | FireFlow |
> |-------------------------|:-----:|:----:|:---------:|:--------:|
> | `--guidance`            |   2   |  2   |     2     |    2     |
> | `--start_layer_index`   |   0   |  0   |     0     |    0     |
> | `--end_layer_index`     |  37   |  37  |    37     |    37    |
> | `--inject`              |   2   |  2   |     2     |    2     |
> | `--reuse_v`             |   1   |  1   |     1     |    1     |
> | `--editing_strategy`    | `replace_v` | `replace_v` | `replace_v` | `replace_v` |
> | `--qkv_ratio`           | `1.0,1.0,1.0` | same | same | same |
>
>
>
>
> **Q3**: Personally, I think the performance improvement of PMI is marginal, especially in the image reconstruction tasks. As illustrated in Figure 1, in the conditional case, the performance of RF-Solver is good enough. Although the unconditional image reconstruction results of PMI in Figure 1 are better, I think it is not necessary to test the performance under this setting because nowadays the description of images are easy to be obtained (for example, by MLLM), which can be served as the condition.
>
> **A3**:
>
> We appreciate the reviewer’s perspective. Our intention is not to treat unconditional reconstruction as a practical task, but to use it as a diagnostic tool to isolate inversion quality. In downstream reconstruction and editing, the final result is influenced by both the inversion trajectory and the prompt quality [1, 2]. Good prompts can mask inversion errors, making conditional evaluation insufficient for identifying whether the inversion itself is accurate. Unconditional reconstruction removes this confounding factor and reveals drift that conditional prompts cannot correct.
> As shown in Figure 1, RF-Solver performs well under conditional prompts but exhibits noticeable drift in the unconditional case, whereas PMI explicitly reduces this accumulated error. We clarify this motivation in **Sec. 3.4** of the revised manuscript.
> Regarding performance magnitude, PMI improves RF-Solver by **1.55** in PSNR and **1.22** in SSIM under the conditional setting, which is substantially larger than the performance gap between RF-Solver and FireFlow. This indicates that PMI provides a significant correction rather than a marginal gain.
>
> [1] "Prompt tuning inversion for text-driven image editing using diffusion models", ICCV 2023
>
> [2] "Source prompt disentangled inversion for boosting image editability with diffusion models", ECCV 2024

---

> ### Author Response · Authors · 2025-11-26
>
> Dear Reviewer,
>
> I hope this message finds you well. We hope we've addressed your concerns. If you have any further questions, please let us know. Your insights are invaluable to us, and we're eager to address any remaining issues to improve our work.
>
> Thank you for your time and effort in reviewing our paper.

---

### Official Review · Reviewer_miCb · 2025-10-30

**Soundness:** 3
**Presentation:** 3
**Contribution:** 2
**Rating:** 4
**Confidence:** 3

**Summary:**

The paper introduces Proximal-Mean Inversion (PMI), a novel, training-free gradient correction method to enhance the stability and accuracy of RF-based generative model inversion. The motivation of this paper is the problem where the existing inversion methods suffer from approximation errors that accumulate across timesteps. PMI addresses this by guiding the perturbed velocity field toward a running average of past velocities using a proximal update. The paper also proposes mimic-CFG, a lightweight velocity correction that interpolates between the current velocity and its projection onto the historical average, balancing editing effectiveness and structural consistency. The proposed method demonstrates good results from the expriments.

**Strengths:**

1. The proposed methods are techically sounds and supported by theoretical proofs.
2. The approach achieves state-of-the-art quality on PIE-Bench with fewer sampling steps and no additional NFEs, accelerating the model.
3. Mimic-CFG provides an efficient guidance mechanism for editing on top of CFG, balancing structural consistency and editing control according to the experiment results.

**Weaknesses:**

1. The performance of the proposed methods seems to be dependent on hyperparameter selection, and could have potential for overcorrection. Over-correction (small $w$) harms both background preservation and editing quality, despite better SSIM/PSNR improvements. Similarly, for the proximal operator parameter $\lambda$ in editing, large values can lead to overcorrection, compromising editing quality.
2. How sensitive is the performance to the hyper-paraetmer across different RF models?
3. While the method is integrated into several solvers (Euler, Heun, RF-Solver, FireFlow), the key ablation studies (on $\lambda$ and $w$) are primarily conducted using one base method (Fig. 3). This limits confidence in how well the optimal hyperparameters and correction strategies generalize across different flow solvers or data distributions.

**Questions:**

see weakness section.

---

> ### Author Response · Authors · 2025-11-22
>
> **Q1**:
> The performance of the proposed methods seems to be dependent on hyperparameter selection, and could have potential for overcorrection. Over-correction (small
> ) harms both background preservation and editing quality, despite better SSIM/PSNR improvements. Similarly, for the proximal operator parameter
>  in editing, large values can lead to overcorrection, compromising editing quality.
>
> **A1**:
> We thank the reviewer for the insightful comment regarding hyperparameter selection. We agree that parameter tuning is important and that overly aggressive correction may lead to suboptimal results. However, as shown in the additional analyses, the performance of PMI is remarkably stable within a broad range of hyperparameters. For example, in **Fig. B.3**, the PSNR varies only from **22.02** to **22.05** (a fluctuation of **0.03**), SSIM varies from **78.86** to **78.94** (a fluctuation of **0.08**), CLIP Similarity (Whole) varies from **21.90** to **21.97** (only **0.07**), and CLIP Similarity (Edited) varies from **19.62** to **19.66** (a fluctuation of **0.04**). These results demonstrate that PMI is not sensitive to hyperparameter changes, and that the apparent risk of overcorrection appears only in extreme parameter choices that are well outside the stable operating region identified by our experiments.
>
>
> **Q2**: How sensitive is the performance to the hyper-paraetmer across different RF models?
>
> **A2**:  Thank you for the question. To evaluate the sensitivity of our method to the hyperparameters across different RF models, we reproduced FireFlow on the SD3.5 model and conducted systematic analyses of both the parameter $\lambda$ and $w$ .
> Since these results are presented as image-based experiments, we provide the full visual comparisons in **Appendix B.4** and **Appendix B.5** (**Fig. B.7, Fig. B.8, Fig. B.9**). These studies show that the effective operating range of both hyperparameters is stable and that the method performs consistently across different RF models (SD3.5).
> We also conduct reconstruction and editing tasks on SD3.5 with Fireflow. The results are summarized in the table below.
>
> ## Reconstruction on SD3.5
>
> | Method            | PSNR ↑ | LPIPS ×10³ ↓ | MSE ×10⁴ ↓ | SSIM ×10² ↑ |
> |-------------------|--------|---------------|--------------|--------------|
> | FireFlow          | 20.91  | 128.16        | 189.79       | 82.32        |
> | FireFlow + Ours   | **20.92**  | **127.89**       | **187.86**      |**82.35**     |
>
> ## Editing on SD3.5
>
> | Method            | PSNR ↑ | LPIPS (×1e3) ↓ | MSE (×1e4) ↓ | SSIM (×1e2) ↑ | Whole ↑ | Edited ↑ |
> |-------------------|--------|----------------|--------------|----------------|---------|----------|
> | FireFlow          | 20.57  | 150.73         | 137.14       | 81.52          | 24.51   | 21.71    |
> | FireFlow + Ours   | **20.67**  | **148.63**         | **135.43**       | **81.76**          | **24.58**   | **21.83**    |
>
> **Q3**: While the method is integrated into several solvers (Euler, Heun, RF-Solver, FireFlow), the key ablation studies (on $\lambda$
>  and $w$
> ) are primarily conducted using one base method (Fig. 3). This limits confidence in how well the optimal hyperparameters and correction strategies generalize across different flow solvers or data distributions.
>
>
>
> **A3**: Thank you for pointing this out. Fig. 3 in the main paper reports the ablation results using FireFlow as the representative solver. To further verify the generality of the hyperparameter choices, we have added additional analyses for Euler, Heun, and RF-Solver.
> These experiments evaluate both the parameter $\lambda$ and $w$ across different solvers and confirm that the effective operating ranges remain stable across the entire spectrum of flow solvers. The full results are provided in **Fig. B.3, Fig. B.4, Fig. 5**, and **Fig. B.6** in the **Appendix B.2** and **Appendix B.3**. Consistent performance confirms that the optimal hyperparameters for PMI are indeed robust and generalize well across different solvers.

---

> ### Author Response · Authors · 2025-11-26
>
> Dear Reviewer,
>
> I hope this message finds you well. We hope we've addressed your concerns. If you have any further questions, please let us know. Your insights are invaluable to us, and we're eager to address any remaining issues tto improve our work.
>
> Thank you for your time and effort in reviewing our paper.

---

> > ### Comment · Reviewer_miCb · 2025-11-27
> >
> > Thank the authors for addressing my concerns, I increased the score to 6.

---

> > > ### Author Response · Authors · 2025-11-27
> > >
> > > Dear Reviewer miCb,
> > >
> > > Thank you for taking the time to review our revised manuscript and for your positive reassessment of our work. We appreciate the increased score.
> > >
> > > Best regards,
> > >
> > > The Authors

---

### Official Review · Reviewer_TwVa · 2025-11-01

**Soundness:** 3
**Presentation:** 3
**Contribution:** 3
**Rating:** 6
**Confidence:** 5

**Summary:**

This paper proposes Proximal-Mean Inversion (PMI) to improve the stability and accuracy of inversion in Flow Matching models. The key idea is to perform a proximal correction of the predicted velocity field by guiding it toward a running average of past velocities, thereby mitigating accumulated approximation errors during inversion. The method also constrains updates within a theoretically derived spherical Gaussian region, ensuring stability in high-dimensional latent spaces. In addition, the authors propose mimic-CFG, a lightweight editing strategy that interpolates between the predicted velocity and its projection on the average direction, effectively balancing structural consistency and editability.

**Strengths:**

* The paper introduces a theoretically grounded proximal correction framework supported by rigorous stability and error analyses..
* The PMI formulation is elegant and practical, addresses the instability and accumulated inversion errors in flow-based generative models.
* Extensive quantitative and qualitative evaluations on PIE-Bench demonstrate consistent improvements across multiple baselines, in both inversion and editing tasks.
* The paper is well written, clearly organized, and easy to follow.

**Weaknesses:**

* Although the focus is on inversion-based editing, it would strengthen the paper to compare against a broader set of diffusion and flow-based editing baselines on PIE-Bench.
* Including the full benchmark results or additional comparisons with diffusion-based methods (e.g., DDIM inversion variants) would provide clearer context.
* Missing relevant recent works such as InfEdit [1], which explores inversion-free diffusion model based editing, and FlowEdit [3] / FlowChef [2], which also employ flow steering without inversion. These could better situate PMI’s contribution in the broader editing landscape.
* The proximal objective (Eq. 9) and its averaging strategy appear empirical. It would be interesting to analyze alternative formulations. For instance, exponential moving averages or adaptive weighting of past velocities.
* While mimic-CFG’s interpolation weight (w = 0.94) is empirically validated, additional discussion on its generalization or task dependency would help establish robustness.

[1] “Inversion-Free Image Editing with Natural Language,” CVPR 2024.

[2] “FlowChef: Steering of Rectified Flow Models for Controlled Generations,” ICCV 2025.

[3] “FlowEdit: Inversion-Free Text-Based Editing Using Pre-Trained Flow Models,” ICCV 2025.

**Questions:**

* The stabilization of inversion and editing velocities is a promising direction. Could this proximal correction mechanism be extended to other tasks, such as inverse problem solving or classifier guidance stabilization (as in [3])?
* Eq. (9) defines a proximal objective with a simple average of velocities. What would happen if this were replaced by an exponential moving average (EMA) or momentum-based scheme? Would it improve convergence or maintain better global consistency over long trajectories?
* Could the authors discuss the potential interaction between PMI and high-order solvers like Adams-Bashforth-Moulton or used by RF-Solver or FireFlow? It would be insightful for readers to understand how the entire workflow evolves.

---

> ### Author Response · Authors · 2025-11-22
>
> **Q1**: The stabilization of inversion and editing velocities is a promising direction. Could this proximal correction mechanism be extended to other tasks, such as inverse problem solving or classifier guidance stabilization (as in [3])?
>
> **A1**: Thank you for the reviewer’s positive recognition of the direction and for raising this insightful question. Our work primarily follows prior studies that apply inversion-stabilization ideas within RF-based inversion and reconstruction, and our formulation is designed around the error structure specific to these tasks [1-3]. At the same time, extending this stabilization perspective to the two settings mentioned by the reviewer, namely inverse problem solving and classifier-guidance stabilization, is indeed very interesting. We appreciate the reviewer for highlighting these potential directions, and we plan to explore whether the idea of stabilizing velocity fields can be effectively adapted to such tasks in future work.
>
> [1] "Taming rectified flow for inversion and editing", ICML 2025
>
> [2] "Fireflow: Fast inversion of rectified flow for image semantic editing", ICML 2025
>
> [3] "Instability in diffusion odes: An explanation for inaccurate image reconstruction", 2025
>
> **Q2**: Eq. (9) defines a proximal objective with a simple average of velocities. What would happen if this were replaced by an exponential moving average (EMA) or momentum-based scheme? Would it improve convergence or maintain better global consistency over long trajectories?
>
> **A2**:
>
> Thank you for the suggestion. We have additionally evaluated an EMA-based velocity averaging scheme and compared it with the original integral averaging strategy used in our method. The results are shown in **Tab. B.1**. Without any parameter tuning, EMA already produces competitive improvements over the baseline FireFlow, we thank you once again for your suggestions. We include it as a variant for discussion in our paper (**Appendix B.7**).
>
> Our final method, which uses the proposed integral-based average, still achieves the best performance across most metrics, except SSIM. This indicates that our velocity formulation provides a more effective control of trajectory stability. We plan to further investigate EMA and other averaging strategies in future work, including a more systematic analysis of their parameterization and integration behavior. The results are summarized in the table below
>
> ## Comparison on different velocity computing schemes
>
> | Method        | Structure Dist (×1e3) ↓ | PSNR ↑ | LPIPS (×1e3) ↓ | MSE (×1e4) ↓ | SSIM (×1e2) ↑ | CLIP Whole ↑ | CLIP Edited ↑ | NFE ↓ |
> |---------------|--------------------------|--------|------------------|----------------|----------------|----------------|-----------------|--------|
> | FireFlow      | 29.39                   | 22.40  | 136.49          | 79.35         | 80.78          | 25.35          | 22.52           | 18     |
> | + EMA         | 28.82                   | 22.74  | 129.36          | 78.59         | **82.97**          | 25.39          | 22.54           | 18     |
> | + Ours        | **28.01**               | **22.79** | **128.55**      | **78.02**     | 81.76       | **25.50**       | **22.68**       | 18     |
>
> **Q3**: Could the authors discuss the potential interaction between PMI and high-order solvers like Adams-Bashforth-Moulton or used by RF-Solver or FireFlow? It would be insightful for readers to understand how the entire workflow evolves.
>
>
> **A3**: We appreciate the reviewer's insightful question regarding the interaction between our proposed approach (PMI) and advanced high-order solvers such as Heun, RF-Solver, and FireFlow. Understanding this integration is crucial for the overall workflow. The detailed algorithms are provided in **Appendix C.5** (**Algorithm 3-8**).
> For higher-order solvers that rely on multiple velocity evaluations (e.g., Heun or Adams methods), the PMI correction is applied to the final integrated velocity before the state update, ensuring the stability mechanism operates on the full-step velocity.

---

> ### Author Response · Authors · 2025-11-22
>
> **Q4**: Including the full benchmark results or additional comparisons with diffusion-based methods (e.g., DDIM inversion variants) would provide clearer context.
>
> **A4**:
>
> Thank you for the suggestion. We additionally evaluated our method on several diffusion-based inversion and editing baselines using SDXL-Turbo. These experiments include DDIM inversion variants and other representative diffusion-model inversion pipelines. The results are summarized in the table below and are further detailed in **Appendix B.7** (**Tab. B.4**).
>
> | Method        | PSNR ↑ | LPIPS(×1e3) ↓ | MSE(×1e4) ↓ | SSIM(×1e2) ↑ | CLIP Whole ↑ | CLIP Edited ↑ |
> |---------------|--------|----------------|--------------|----------------|----------------|-----------------|
> | GNRI          | 21.96  | 126.63         | 93.27        | 75.00          | 24.90          | 21.98           |
> | Turbo-Edit   | 22.26  | 107.30         | 99.96        | 80.05          | 25.85          | 22.74           |
> | Direct Inv    | 22.46  | 105.53         | 79.96        | 80.22          | 25.64          | 22.68           |
> | DDIM         | 22.00  | 118.56         | 90.16        | 76.00          | 24.38          | 21.48           |
> | + Ours | **22.49**(+0.49) | **105.17** (−13.9) | **84.01** (−6.15) | **80.27** (+4.27) | **25.62** (+1.24) | **22.70** (+1.22) |
>
> Across all tested methods on SDXL-Turbo, PMI consistently reduces inversion drift and improves reconstruction quality, demonstrating that our correction strategy generalizes well beyond RF-based solvers and remains effective when applied to diffusion-based methods.
>
> **Q5**: Missing relevant recent works such as InfEdit, which explores inversion-free diffusion model based editing, and FlowEdit / FlowChef, which also employ flow steering without inversion. These could better situate PMI’s contribution in the broader editing landscape.
>
> **A5**:
>
> We thank the reviewer for highlighting these important, relevant works. We agree that InfEdit, FlowEdit, and FlowChef are crucial for situating our work within the broader editing landscape, especially considering their focus on inversion-free or flow-steering approaches. We have now added a detailed discussion of these papers in **Sec. 2 (Editing)** of the revised manuscript.
>
> **Q6**: While mimic-CFG’s interpolation weight (w = 0.94) is empirically validated, additional discussion on its generalization or task dependency would help establish robustness.
>
> **A6**:
>
> Thank you for the helpful comment. To address this point, we provide an extended analysis of different interpolation weights in **Appendix B.3 (Fig. B.6)**, evaluated across Euler, Heun, and RF-Solver. Consistent performance confirms that the optimal hyperparameters for mimic-CFG are indeed robust and generalize well across different solvers.

---

> ### Author Response · Authors · 2025-11-26
>
> Dear Reviewer,
>
> I hope this message finds you well. We hope we've addressed your concerns. If you have any further questions, please let us know. Your insights are invaluable to us, and we're eager to address any remaining issues to improve our work.
>
> Thank you for your time and effort in reviewing our paper.

---

### Meta-Review · Area_Chair_paq4 · 2026-01-06

**Summary:**

This paper proposes PMI, a training-free correction method for improving inversion and editing stability in rectified-flow models, along with a lightweight mimic-CFG strategy for editing. Reviewers generally agreed that the problem being addressed is real and relevant, the method is technically sound, and the paper is carefully written. Several reviewers appreciated the clean formulation, the fact that the method adds no extra NFEs, and the consistent improvements across multiple solvers and benchmarks. The rebuttal was thorough and resolved many concrete concerns about ablations, baseline settings, and hyperparameter sensitivity, and at least one reviewer explicitly raised their score after these clarifications.

**Reviewer Concerns:**

At the same time, there were persistent reservations that prevent this from being a strong accept from AC's view. A recurring concern is that the conceptual contribution is modest. The core update ultimately reduces to a simple, first-order correction guided by a running mean, and some reviewers felt that the “proximal” framing and parts of the theoretical analysis overstate the depth of what is essentially a well-motivated heuristic. Even after revisions, parts of the theory (e.g., error bounds and stability constants) were seen as loosely connected to the empirical gains rather than truly explanatory.

Another concern is the size and significance of the improvements. While the gains are consistent, several reviewers viewed them as incremental, especially for conditional reconstruction where existing solvers already perform well. The editing results, although improved in terms of background consistency and CLIP scores, remain somewhat subtle, and the evaluation still relies heavily on proxy metrics rather than strong evidence of clearly better semantic edits.

Overall, the paper sits right on the borderline I would say. It is solid, and practically useful, and the authors engaged seriously with reviewer feedback. AC leans toward a weak accept, but I would also be comfortable with a reject decision if space is tight.

**Reviewer Scores:**

Before the rebuttal stage, the overall score was 6-4-4-6, and we believe it would likely have improved to 6-6-4-6

---

### Decision · Program_Chairs · 2026-01-26

Accept (Poster)